# Is Your Model Really A Good Math Reasoner? Evaluating Mathematical Reasoning with Checklist

**Zihao Zhou**[12*]   **Shudong Liu**[3*]   **Maizhen Ning**[126]   **Wei Liu**[4]   **Jindong Wang**[5]
**Derek F. Wong**[3]   **Xiaowei Huang**[2]   **Qiufeng Wang**[1†]   **Kaizhu Huang**[6]
[1]Xi'an Jiaotong-liverpool University   [2]University of Liverpool   [3]University of Macau
[4]HKUST   [5]William & Mary   [6]Duke Kunshan University
**https://mathcheck.github.io/**

## Abstract

Exceptional mathematical reasoning ability is one of the key features that demonstrate the power of large language models (LLMs). How to comprehensively define and evaluate the mathematical abilities of LLMs, and even reflect the user experience in real-world scenarios, has emerged as a critical issue. Current benchmarks predominantly concentrate on problem-solving capabilities, presenting a substantial risk of model overfitting and fails to accurately measure the genuine mathematical reasoning abilities. In this paper, we argue that if a model really understands a problem, it should be robustly and readily applied across a diverse array of tasks. To this end, we introduce MATHCHECK, a well-designed checklist for testing task generalization and reasoning robustness, as well as an automatic tool to generate checklists efficiently. MATHCHECK includes multiple mathematical reasoning tasks and robustness tests to facilitate a comprehensive evaluation of both mathematical reasoning ability and behavior testing. Utilizing MATHCHECK, we develop **MATHCHECK-GSM** and **MATHCHECK-GEO** to assess mathematical textual reasoning and multi-modal reasoning capabilities, respectively, serving as upgraded versions of benchmarks including GSM8k, GeoQA, UniGeo, and Geometry3K. We adopt MATHCHECK-GSM and MATHCHECK-GEO to evaluate over 26 LLMs and 17 multi-modal LLMs, assessing their comprehensive mathematical reasoning abilities. Our results demonstrate that while frontier LLMs like GPT-4o continue to excel in various abilities on the checklist, many other model families exhibit a significant decline. Further experiments indicate that, compared to traditional math benchmarks, MATHCHECK better reflects true mathematical abilities and represents mathematical intelligence more linearly, thereby supporting our design. Using MATHCHECK, we can also efficiently conduct informative behavior analysis to deeply investigate models. Finally, we show that our proposed checklist paradigm can easily extend to other reasoning tasks for their comprehensive evaluation.

## 1 Introduction

The AI community has been placing significant emphasis on mathematical reasoning as a means to explore the upper limits of intelligence in large language models (LLMs) (Achiam et al., 2023; Team et al., 2023; Meta, 2024; Jiang et al., 2024; Wei et al., 2022; Trinh et al., 2024; Romera-Paredes et al., 2024) and multi-modal large language models (MLLMs) (OpenAI, 2024c; Lu et al., 2023). A large number of efforts have been made on how to enhance (M)LLMs' mathematical reasoning abilities. In pre-training, Wang et al. (2023d); Shao et al. (2024); Lin et al. (2024); Zhang et al. (2024c) studied the impact of the quality of mathematical corpus; in post-training, Yue et al. (2023); Yu et al. (2023); Li et al. (2024a) augmented a huge number of synthetic data, and then developed

---

*Equal contribution. Email: `zihao.zhou@liverpool.ac.uk`; `nlp2ct.shudong@gmail.com`
†Corresponding author. Email: `Qiufeng.Wang@xjtlu.edu.cn`

|  | Problem Solving | Answerable Judging | Outcome Judging | Process Judging |
|---|---|---|---|---|
| **Original Problem** | A robe takes 2 bolts of blue fiber and half that much white fiber. How many bolts in total does it take?
"answer": 3.0 | A robe **takes bolts** of blue fiber and half that much white fiber. How many bolts in total does it take?
"answer": Unanswerable ⭐**Seed Data** | A robe takes 2 bolts of blue fiber ... How many bolts in total does it take?
"solution": Step 1: 2 bolts of blue fiber...The answer is **4 bolts** in total.
"answer": Incorrect | A robe takes 2 bolts of blue fiber ... How many bolts in total does it take?
"solution": Step 1: Identify the amount ... Step 3: **Multiply** the bolts of blue and white fiber together to find the total number of bolts. The answer is 2 bolts.
"answer": Step 3 |
| **Problem Understanding** | To make a robe, you need 2 bolts of blue fiber and half as many bolts of white fiber compared to blue. What is the total number of bolts required for the robe?
"answer": 3.0 | To make a robe, you **need bolts** of blue fiber and half as many bolts of white fiber compared to blue. What is the total number of bolts required for the robe?
"answer": Unanswerable | To make a robe, you need 2 bolts ... What is the total number of bolts required for the robe?
"solution": Step 1: Calculate the number of blue bolts... So, 2 (blue)+ 1 (white) = 3.The answer is **3**.
"answer": Correct | To make a robe, you need 2 bolts is the total number of bolts required for the robe?
"solution": Step 1: ... Step 2: Determine the number of white bolts, which **as many as blue bolts**. ... The answer is 4.
"answer": Step 2 |
| **Irrelevant Disturbance** | A tailor is crafting a luxurious robe. The design requires 2 bolts of blue fiber and half that amount of white fiber. To add grandeur, the tailor also considered using 3 bolts of golden thread from the sun's rays, but eventually decided it would be too gaudy for the ceremony. How many bolts in total are needed for the robe, disregarding the golden thread?
"answer": 3.0 | A tailor is crafting a luxurious robe. The design requires 2 bolts of blue fiber and half that amount of white fiber. ... How many bolts in total are needed for the robe, disregarding the golden thread?
"answer": Answerable | A tailor is crafting a luxurious robe. The design requires 2 bolts of blue fiber and half that amount of white fiber. ... How many bolts in total are needed for the robe, disregarding the golden thread?
"solution": Step 1: Calculate the amount of blue fiber. The design requires ... The answer is: **300 yards**.
"answer": Incorrect | A tailor is crafting a luxurious robe. The design requires 2 bolts of blue fiber and half that amount of white fiber. ... How many bolts in total are needed for the robe, disregarding the golden thread?
"solution": Step 1: ... Step 2: Calculate the amount of white fiber required, which is double the blue fiber amount, so **2 bolts * 2 = 4 bolts**. Step 3: ... The answer is 6 bolts.
"answer": Step 2 |
| **Scenario Understanding** | A robe takes x bolts of blue fiber and half that much white fiber. It takes 3 bolts in total. What is the value of unknown variable x?
"answer": 2.0 | A robe takes x bolts of blue fiber and **fewer** white fiber. It takes 3 bolts in total. What is the value of unknown variable x?
"answer": Unanswerable | A robe takes x bolts of blue fiber and half that ... What is the value of unknown variable x?
"solution": Step 1: Let's say the value of x is ... The answer is **2**.
"answer": Correct | A robe takes x bolts of blue fiber and half that ... What is the value of unknown variable x?
"solution": Step 1: Let's ... Step 3: To find out how many bolts of fiber are needed in total, the equation should be **x − 0.5x = 3** ... The answer is x equals 6.
"answer": Step 3 |

*Reasoning Robustness* (vertical axis) — *Task Generalization* (horizontal axis)

Figure 1: Overview of MATHCHECK design. The horizontal axis examines the task generalization of four math tasks while the vertical axis examines the reasoning robustness through four problem varieties. All data are generated from seed data, which is also from a mainstream benchmark dataset.

supervised fine-tuning (SFT) for math problem-solving. Recently, Luong et al. (2024) and Sun et al. (2024b) explored variants of reinforcement learning (RL) for further improvements.

To guarantee the high mathematical reasoning ability has been reached, it is crucial to fairly evaluate models' performance. Current mainstream methods rely on the performance across math problem-solving tasks of varying difficulty levels, such as GSM8k (Cobbe et al., 2021) of elementary level, MATH (Hendrycks et al., 2021) of high school level, and TheromQA (Chen et al., 2023a) of university level. Recently, some mathematical datasets that are more challenging, diverse, and multi-modal have been proposed to enhance the mathematical evaluation (He et al., 2024; Liu et al., 2024c; Lu et al., 2023; Zhang et al., 2024b). However, these current evaluation methods focus on *individual* tasks (most of which are problem-solving) and robustness tests for each problem. In other words, they do not provide comprehensive guidance on whether LLMs really achieve mathematical reasoning ability. In this paper, we argue that: *if a model really understands a problem, it should work robustly across various tasks about this problem.* Therefore, it is necessary to evaluate models by multi-tasks with diverse robustness test. Through such investigation, the real reasoning ability of a model can be comprehensively evaluated. As a result, we can also perform detailed behavior tests on models (Ribeiro et al., 2020).

Drawing motivations from this insight, we introduce **MATHCHECK**, a well-designed checklist for testing task generalization and reasoning robustness. MATHCHECK includes general mathematical reasoning tasks and diverse robustness testing types to facilitate a comprehensive evaluation of mathematical reasoning ability and reasoning behavior testing. As shown in Figure 1, horizontally, we examine the task generalization including problem solving, answerable judging, outcome judging, and process judging. Vertically, we test the reasoning robustness through the original problem and its three robustness variants consisting of problem understanding, irrelevant disturbance, and scenario understanding. The data of each cell in the checklist corresponds to a specific type of robustness test and task form. To facilitate the construction of checklist, we propose an (M)LLMs-driven generation framework to automatically generate this data. Figure 2 illustrates the MATHCHECK data collection

process, where the seed solving problem is firstly rewritten to its robustness problems, next all generated solving data are utilized to construct other task forms.

Utilizing MATHCHECK, we propose **MATHCHECK-GSM**, a MATHCHECK dataset generated from GSM8k (Cobbe et al., 2021). It contains a total of 3,096 high-quality samples consisting of 129 groups checklist matrix, which can be used to evaluate mathematical textual reasoning ability comprehensively. Besides, acknowledging the community's focus on multi-modal reasoning capabilities, we further propose **MATHCHECK-GEO** to evaluate the multi-modal geometry reasoning ability. Generated from GeoQA (Chen et al., 2021), UniGeo (Chen et al., 2022), and Geometry3K (Lu et al., 2021), it contains a total of 1,440 samples with a checklist matrix of 60 groups. It is noteworthy that the construction pipeline of MATHCHECK can be applied to most mathematical datasets to dynamically establish a comprehensive and flexible evaluation benchmark, thereby mitigating data contamination (Zhou et al., 2023a; Zhu et al., 2024a;b).

We conduct extensive experiments on 26 LLMs and 17 MLLMs including different scales, API-base and open source, generalist and mathematical models. We find that frontier LLMs like GPT-4o continue to achieve superior performance in our MATHCHECK, but many other model families exhibit a significant decline. Further experiments indicate that compared to solving original problems which is the paradigm of mainstream benchmark, our MATHCHECK evaluation aligns more accurately with the genuine mathematical reasoning ability of the model. Utilizing MATHCHECK, we extensively analyze the models' behaviors including training on massive solving data, reasoning consistency, performance on different complexity problems and applying different prompting technologies. Finally, we show the potential of applying MATHCHECK paradigm to other reasoning tasks such as commonsense reasoning and code generation, promoting more comprehensive evaluation of reasoning ability.

## 2 MATHCHECK

MATHCHECK is a well-designed checklist that includes general mathematical reasoning tasks and diverse robustness testing types for comprehensive evaluation, as well as a tool to automatically generate a large number of test cases in the manner of checklist. In our checklist, various mathematical tasks are arranged in rows to assess task generalization, whereas diverse variants of mathematical problems are placed in columns to evaluate reasoning robustness. We will elaborate on the task types in Section 2.1, problem variants in Section 2.2, and how we construct checklist data in Section 2.3.

### 2.1 TASK GENERALIZATION

Testing models across different tasks on the same domain not only offers a comprehensive and profound evaluation of their capabilities (Frank, 2023) but also caters to the practical demands and complexities of real-world applications (Ji et al., 2023). In MATHCHECK, we incorporate four math tasks including Problem Solving, Answerable Judging, Outcome Judging, and Process Judging.

**Problem Solving.** In this task, we ask the model to solve a given math problem. As the most widely used method to test mathematical reasoning ability in contemporary research (Cobbe et al., 2021; Hendrycks et al., 2021), it necessitates the model to analyze the problem, recall and apply appropriate math knowledge, and finally conclude reasoning results.

**Answerable Judging.** Given a math problem, models need to determine whether the problem provides sufficient information to answer the question. This task requires the model to analyze the question, then identify the essential conditions required for solving this question, subsequently verify whether these conditions are provided within the problem statement. Previous works utilized it to examine whether the model is a reasoner with critical thinking instead of a random parrot (Li et al., 2024b; Sun et al., 2024a; Ma et al., 2024).

**Outcome Judging.** Given a math problem and one of its solutions, let the model determine whether the final answer of the given solution is correct. Outcome-Judging is a coarse-grained judgment of solutions since the model only focuses on the correctness of the final answer. Researchers often apply the outcome-judging ability of models to verify the correctness of augmented data (Tang et al., 2024) and provide outcome rewards in reinforcement learning (Luong et al., 2024).

**Process Judging.** Given a math problem along with its wrong solution, the model is required to identify the step where the errors begin. Compared with the outcome-judging, the process-judging

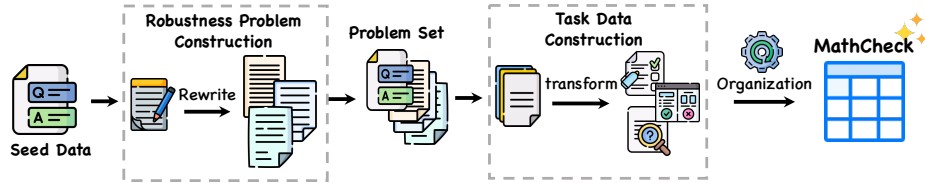

Figure 2: MATHCHECK generation pipeline.

task is a more fine-grained judgment on the solution, which demands the model to judge step by step until the wrong step is located. It can help to debug the given wrong solution.

## 2.2 REASONING ROBUSTNESS

A model that truly understands the inherent mathematical logic of a problem will exhibit reasoning robustness to diverse variations of this problem (Stolfo et al., 2023). Motivated by this, we utilize four problem forms including the original problem and its three rewritten variants to examine the reasoning robustness of models.

**Original Problem.** It is the seed problem of other reasoning robustness variants. At a minimum functionality test, it can check whether the model has the basic mathematical capabilities when no modifications have been made.

**Problem Understanding.** It refers to transforming the original problem into a new one that uses different wording or different sentence structures but does not change the mathematical logic of its original version (Patel et al., 2021; Zhou et al., 2024; Li et al., 2024b). It pays more attention to semantic robustness, and aims to examine whether models can correctly reason when dealing with different descriptions of the same mathematical logic.

**Irrelevant Disturbance.** It refers to inserting irrelevant conditions that are related to the topic of the original question, but have no impact on the final answer. Previous studies have disclosed that large language models are easily distracted by such perturbations (Shi et al., 2023). It needs the model to distinguish which conditions are necessary and which are irrelevant to the problem.

**Scenario Understanding.** When models comprehend the scenario of a math problem and its underlying logic, they should be able to solve other questions within that scenario (Liu et al., 2021; Yu et al., 2023; Zhou et al., 2023b). Therefore, we alter the original question to evaluate whether a model has a comprehensive understanding of the scenario. For example, as shown in Figure 1, we ask the question "the number of blue bolts" instead of "the number of total bolts".

## 2.3 CHECKLIST CONSTRUCTION

Creating MATHCHECK data is a labor-intensive and time-consuming process. The advent of LLMs has introduced a new level of flexibility and quality to generate mathematical content (Norberg et al., 2023; Li et al., 2024b). Therefore, we employ (M)LLMs (e.g., GPT-4-Turbo in our experiments) as engines to automatically generate our MATHCHECK data. The data construction pipeline is shown in Figure 2. Users first assemble a collection of math problems with labels as seed data. Second, (M)LLMs initially rewrite these problems into their robustness varieties to make up the robustness problem set. Third, each problem in this set will be extended to construct multiple mathematical tasks about this problem. Finally, all data are manually checked to form MATHCHECK dataset correctly.

Based on the seed data, we automatically generate another three robustness problems as shown in the first column of Figure 1. *Problem Understanding* and *Irrelevant Disturbance* are the tasks of rewriting problems without altering the final answer. Hence, we prompt the model to rewrite our math problems while maintaining the original answer. For *Scenario Understanding*, we first extract a variable from the problem as a new answer, then prompt the model to change the question based on the extracted variable. Once we obtain the four robustness reasoning problems of the solving task, we rewrite them respectively to construct multiple tasks, including *Answerable Judging*, *Outcome Judging* and *Process Judging* as shown in the corresponding row of Figure 1. For the *Answerable*

*Judging* task, we prompt the model to eliminate a condition from the original problem which is crucial for solving it to obtain an unanswerable problem. For *Outcome Judging* task, we ask the model to solve the problem and acquire candidate solutions, then these solutions are labeled (Correct or Incorrect) according to the final answer. For *Process Judging* task, we apply the solution rewritten ability of (M)LLMs to construct process-judging data. Specifically, given a problem along with its correct solution, we prompt the model to make mistakes from the given steps and results in a wrong answer. In such a way, we can get a wrong solution while its mistake steps remain simultaneously. All of our prompts are listed in Appendix F.2.

## 3 EXPERIMENTS

### 3.1 DATASETS

We use MATHCHECK to comprehensively measure the mathematical reasoning ability across textual and multi-modal settings. Consequently, two benchmarks MATHCHECK-GSM and MATHCHECK-GEO are introduced.

**MATHCHECK-GSM** is a MATHCHECK dataset generated from GSM8k (Cobbe et al., 2021). We choose GSM8k as the seed benchmark since (1) it is most widely used for evaluating mathematical textual reasoning capability. (2) we aim to determine whether advanced models are genuinely capable of reasoning at the grade school level. We first collect a test-mini set of GSM8k, which includes 129 problems sampled evenly according to the difficulty[1]. Subsequently, we generate 129 MATHCHECK style groups, totaling 3,096 high-quality samples by MATHCHECK. It can be used to evaluate the real mathematical reasoning ability of LLMs on GSM8k-level problems. A group of MATHCHECK-GSM case problems are listed in Appendix G.1.

**MATHCHECK-GEO** is a dataset for geometry problems, which is the representative task for evaluating multi-modal reasoning capability. First, we collect seed geometry problems from GeoQA (Chen et al., 2021), UniGeo (Chen et al., 2022), and Geometry3K (Lu et al., 2021), containing 60 problems in both English and Chinese. Subsequently, we generate 60 MATHCHECK style groups, totaling 1,440 high-quality samples. Notably, this is the first geometry problem dataset involving answerable, outcome, and process judgment tasks. MATHCHECK-GEO gives research community a harder and multi-modal MATHCHECK style dataset, as well as showing the extensibility of MATHCHECK. A group of MATHCHECK-GEO case problems are shown in Appendix G.2.

All datasets are checked with meticulous manual validation to ensure high quality and reliability. To this end, we recruited three graduate students who underwent training tailored to the requirements of our research. This rigorous verification process not only enhances the quality of our data but also reinforces the validity of our findings. Finally, our automatic data generation pipeline can achieve an average pass rate of 84.61% (Appendix C.2). The detailed data statistics and quality discussion of our checklist are reported in Appendix C.

### 3.2 EXPERIMENTAL SETUP

To systematically benchmark the mathematical reasoning capabilities of existing LLMs, we include a comprehensive evaluation of 43 models, comprising 26 LLMs and 17 MLLMs. These models are principally divided into two categories: generalist models encompassing both API-based commercial LLMs and open-sourced LLMs (large and small scale), and specialized mathematical models. We use the F1 metric for Outcome Judging and Answerable Judging tasks, and the Acc metric for the other two tasks. The list of selected models and details of evaluation setup can be found in Appendix D.

### 3.3 MAIN RESULTS

Tables 1 and 2 illustrate the performance of various models on the MATHCHECK-GSM and MATHCHECK-GEO, respectively. The leftmost column represents the average performance across all tasks and all question variants. The middle four columns detail the performance on various mathematical reasoning tasks, while the right four columns display performance across different question variants. Consequently, each model is represented by a $4 \times 4$ checklist table, which showcases the

---

[1]We define the difficulty according to the number of reasoning steps of its answers (2 steps to 8 steps)

Table 1: Model performance on MATHCHECK-GSM. **PS**: **P**roblem **S**olving, **AJ**: **A**nswerable **J**udging, **OJ**: **O**utcome **J**udging, **PJ**: **P**rocess **J**udging, **OP**: **O**rigianl **P**roblem, **PU**: **P**roblem **U**nderstanding, **ID**: **I**rrelevant **D**isturbance, **SU**: **S**cenario **U**nderstanding. Each score is the average score of related units. For example, 'All' means all units, 'PS' includes solving units on four problem types, 'OP' includes original problems on four tasks units.

| Models | All | PS | AJ | OJ | PJ | OP | PU | ID | SU |
|---|---|---|---|---|---|---|---|---|---|
| *Generalist Models* | | | | | | | | | |
| O1-preview | 93.2 | 91.3 | 94.0 | 93.2 | 94.1 | 95.6 | 93.4 | 90.5 | 93.1 |
| O1-mini | 92.7 | 93.6 | 95.0 | 88.9 | 93.6 | 95.5 | 94.2 | 91.0 | 90.5 |
| GPT-4o | 92.0 | 95.0 | 95.0 | 90.1 | 87.8 | 94.6 | 91.6 | 92.0 | 89.6 |
| GPT-4o-mini | 87.2 | 90.1 | 89.6 | 88.6 | 80.4 | 88.9 | 89.4 | 85.6 | 85.1 |
| GPT-4-Turbo-20240409 | 90.9 | 93.8 | 95.9 | 87.8 | 86.0 | 93.8 | 90.4 | 90.8 | 88.6 |
| GPT-3.5-Turbo | 61.4 | 73.5 | 64.3 | 48.3 | 59.5 | 65.4 | 64.6 | 60.1 | 55.4 |
| Gemini-1.5-Pro | 86.3 | 88.6 | 89.5 | 87.6 | 75.0 | 88.0 | 90.2 | 85.0 | 82.0 |
| Claude-3.5-sonnet-20240620 | 90.2 | 94.8 | 95.3 | 90.9 | 79.9 | 92.5 | 92.1 | 89.9 | 86.3 |
| Claude-3-opus-20240229 | 83.5 | 81.6 | 92.0 | 78.7 | 81.8 | 86.3 | 85.6 | 81.9 | 80.3 |
| Claude-3-sonnet-20240229 | 75.0 | 77.9 | 88.9 | 65.1 | 68.0 | 76.5 | 77.8 | 73.7 | 71.9 |
| Claude-3-haiku-20240229 | 57.5 | 79.7 | 49.9 | 44.3 | 56.0 | 61.9 | 62.4 | 55.9 | 49.6 |
| Llama-3.1-70B-Instruct | 90.5 | 95.2 | 95.3 | 89.4 | 82.2 | 93.3 | 91.2 | 89.8 | 87.7 |
| Llama-3-70B-Instruct | 84.7 | 90.1 | 87.5 | 84.6 | 76.7 | 87.7 | 86.7 | 84.7 | 79.9 |
| DeepSeek V2 | 82.2 | 86.8 | 82.6 | 82.5 | 76.9 | 85.1 | 84.4 | 83.5 | 75.9 |
| Mixtral 8 x 7B-Instruct | 59.9 | 56.0 | 58.1 | 63.9 | 61.6 | 62.8 | 61.5 | 58.8 | 56.4 |
| Mixtral 8 x 7B-Base | 44.7 | 40.9 | 50.8 | 51.8 | 35.3 | 50.6 | 47.8 | 41.2 | 39.1 |
| Qwen1.5-72B-Chat | 50.6 | 71.1 | 64.2 | 31.9 | 35.1 | 57.0 | 51.1 | 43.6 | 50.6 |
| Phi-3-Medium-4K-Instruct | 72.0 | 89.7 | 70.8 | 63.2 | 64.1 | 77.6 | 78.7 | 71.1 | 60.4 |
| Phi-3-Mini-4K-Instruct | 64.1 | 71.3 | 64.5 | 62.9 | 57.6 | 68.5 | 66.6 | 61.2 | 60.0 |
| Llama-3.1-8B-Instruct | 71.0 | 76.9 | 65.8 | 77.2 | 64.0 | 74.6 | 73.6 | 66.0 | 69.6 |
| Llama-3-8B-Instruct | 64.2 | 68.6 | 61.4 | 64.9 | 61.8 | 67.8 | 68.8 | 62.9 | 57.1 |
| ChatGLM3-6B | 36.5 | 32.6 | 41.7 | 50.1 | 21.7 | 39.7 | 35.9 | 31.3 | 39.1 |
| *Mathematical Models* | | | | | | | | | |
| DeepSeek-Math-7B-RL | 50.7 | 79.5 | 50.0 | 45.1 | 28.1 | 53.3 | 51.2 | 47.5 | 50.6 |
| DeepSeek-Math-7B-Instruct | 50.2 | 70.0 | 64.8 | 40.4 | 25.8 | 51.6 | 54.4 | 45.8 | 49.2 |
| DeepSeek-Math-7B-Base | 44.0 | 49.8 | 51.5 | 44.0 | 30.8 | 49.0 | 46.0 | 37.0 | 44.1 |
| MetaMath-LLama2-70B | 45.7 | 70.0 | 35.7 | 45.3 | 31.6 | 49.9 | 51.5 | 43.4 | 37.8 |

Table 2: Model performance on MATHCHECK-GEO.

| Models | All | PS | AJ | OJ | PJ | OP | PU | ID | SU |
|---|---|---|---|---|---|---|---|---|---|
| *Generalist Models* | | | | | | | | | |
| GPT-4o | 65.3 | 57.5 | 75.5 | 69.5 | 58.8 | 65.2 | 67.0 | 64.3 | 64.8 |
| GPT-4o-mini | 59.0 | 50.8 | 69.8 | 61.4 | 53.8 | 61.9 | 62.0 | 54.1 | 57.8 |
| GPT-4-Turbo-20240409 | 61.7 | 51.3 | 72.3 | 64.0 | 59.2 | 63.2 | 62.9 | 61.7 | 58.9 |
| GPT-4-Vision-Preview | 60.0 | 46.7 | 71.1 | 63.6 | 58.8 | 59.3 | 62.8 | 57.8 | 60.2 |
| Gemini-1.5-Pro | 58.7 | 47.5 | 67.4 | 55.0 | 64.6 | 62.3 | 58.6 | 57.1 | 56.9 |
| Gemini-1.5-Flash | 56.8 | 45.0 | 75.1 | 50.6 | 56.7 | 56.8 | 59.7 | 53.8 | 57.1 |
| Claude-3.5-sonnet-20240620 | 58.7 | 54.2 | 71.0 | 53.0 | 56.7 | 59.9 | 63.8 | 54.3 | 56.8 |
| Claude-3-opus-20240229 | 47.2 | 34.2 | 60.6 | 46.7 | 47.5 | 47.2 | 49.1 | 42.4 | 50.2 |
| Claude-3-sonnet-20240229 | 49.9 | 35.8 | 59.0 | 51.6 | 52.9 | 51.2 | 53.0 | 44.7 | 50.4 |
| Claude-3-haiku-20240307 | 36.7 | 27.9 | 41.3 | 41.7 | 35.8 | 39.2 | 38.8 | 33.3 | 35.4 |
| QWen2-VL-72B-Instruct | 61.4 | 60.0 | 53.1 | 61.3 | 71.3 | 69.0 | 62.4 | 58.0 | 56.4 |
| QWen2-VL-7B-Instruct | 42.1 | 35.8 | 49.4 | 46.4 | 36.7 | 40.9 | 45.6 | 41.7 | 40.0 |
| InternVL-1.5-Chat | 37.6 | 22.1 | 54.9 | 46.8 | 26.7 | 42.9 | 34.8 | 37.3 | 35.5 |
| MiniCPM-Llama3-V-2.5 | 37.3 | 37.5 | 38.1 | 45.0 | 28.8 | 37.4 | 45.0 | 35.2 | 31.6 |
| LLaVA-1.6-Mistral-7B-Instruct | 31.8 | 10.0 | 38.8 | 51.2 | 27.1 | 33.8 | 35.5 | 28.4 | 29.2 |
| Phi-3-Vision-128k-Instruct | 29.6 | 12.9 | 35.0 | 48.6 | 22.9 | 32.6 | 31.8 | 28.2 | 26.0 |
| CogVLM2-Llama3-Chat-19B | 24.6 | 7.9 | 26.4 | 46.3 | 17.9 | 27.2 | 28.0 | 22.4 | 20.9 |

model's performance in various dimensions. The details of all checklist tables are further elaborated in Appendix A and B.

On MATHCHECK-GSM (Table 1), O1-preview and O1-mini exhibit outstanding performance with impressive overall score of 93.2 and 92.7, demonstrates strong effect of extending reasoning thought exploration. GPT-4o is closely followed with a score of 92.0 and demonstrates top performance on the problem solving task and irrelevant disturbance variants. These results indicate that strong foundational models still possess formidable and robust performance across a variety of mathematical reasoning tasks. Among the open-source LLMs, LlaMa-3.1-70B-Instruct achieves the highest score of 90.5 and performs excellently across a range of tasks and problem variants. Its performance has significantly improved compared to LLaMA-3 version and surpasses that of GPT-4o-mini. Besides, Qwen1.5-72B-Chat underperforms in tasks other than problem solving, which we suspect is due to its special optimization of the solving task. This phenomenon is also observed across all math-customized models, which tend to be trained on similar mathematical problems and problem-solving processes, resulting in a relatively narrow scope of reasoning capabilities.

On MATHCHECK-GEO (Table 2), GPT-4o demonstrates the best performance, achieving a top score of 65.3 in the All category. The performance of GPT4-turbo-20240409 and GPT4-Vision-Preview is similar, reaching scores of 61.7 and 60.0, respectively. In particular, the performance of Claude-3-sonnet is slightly superior in visual contexts compared to that of its larger counterpart, Claude-3-opus. Among the open-source MLLMs, the large-size MLLMs demonstrate surprisingly strong performance, with Qwen-VL-70B attaining 60.4 over the GPT-4-Vision-Preview. However, the most of small-size MLLMs exhibited poor performance especially in probelm solving, which suggests that the multi-modal reasoning capabilities of open-source small-size open-source MLLMs still have significant room for improvement.

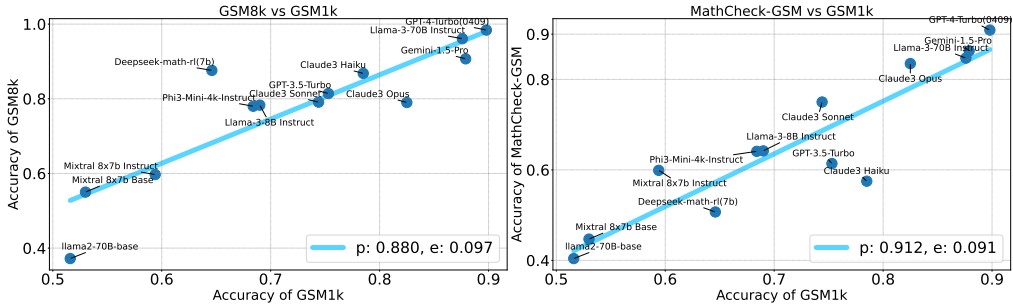

Figure 3: Correlation with GSM1k (Zhang et al., 2024a), a dataset that reflects real mathematical reasoning ability. $p$ and $e$ represent the Pearson Correlation Coefficient, and Root Mean Square Error.

## 3.4 MATHCHECK REPRESENTS MATHEMATICAL INTELLIGENCE MORE LINEARLY

One desiderata of a good mathematical benchmark is to reflect real mathematical intelligence perfectly. We follow previous works (Zhang et al., 2024a; Huang et al., 2024a) to assess "intelligence" from practical standpoints and use performance on private data (Zhang et al., 2024a) and compression efficiency (Du et al., 2024; Huang et al., 2024a) as surrogates to assess the genuine mathematical abilities of models. By examining the correlation between MATHCHECK and these surrogates, we can verify whether our design effectively reflects mathematical intelligence, and how it compares to traditional benchmarks.

**Correlation with Private Data.** Unlike traditional open-sourced benchmarks, private data is less likely to be contaminated or overfitted, making it an appropriate proxy of genuine mathematical intelligence. We adopt GSM1k (Zhang et al., 2024a), a new private GSM8k-level dataset, to measure the real mathematical reasoning of models. We compare the correlation of model performance between GSM1k and MATHCHECK-GSM/GSM8k. As shown in Figure 3, the left part illustrates the correlation between GSM8k and GSM1k. It reveals that most LLMs achieve scores up to 80% on GSM8k, with scores concentrated in the top half of the graph. However, on GSM1k, the scores are evenly distributed, indicating that some LLMs, such as deepseek-math-7B-RL, have inflated scores on GSM8k. This suggests that the GSM8k score is not a reliable benchmark for assessing the true mathematical reasoning ability of the models. In the right sub-figure, MATHCHECK-GSM and GSM1k display a good positive correlation, and some models that do not perform well on GSM1k can be detected by MATHCHECK-GSM. By comparing the Pearson correlation coefficient and the

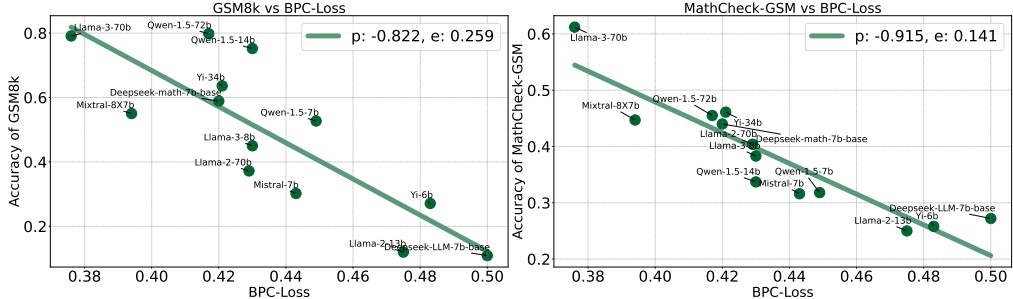

Figure 4: Performance correlation with BPC-loss, which reflects compression efficiency (Huang et al., 2024a). The lower BPC-loss represents the higher compression efficiency.

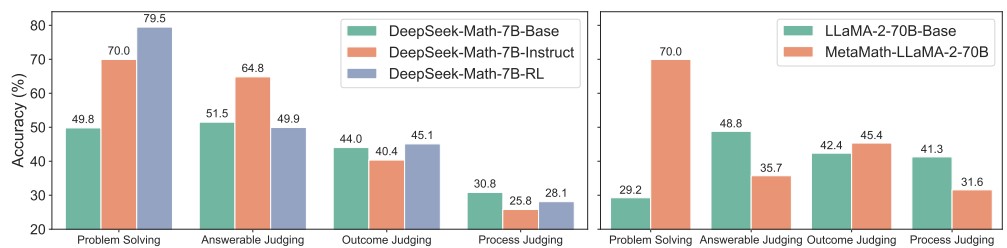

Figure 5: Behavior of mathematical models trained on massive solving data.

root mean square error, it shows that MATHCHECK has a higher correlation coefficient with GSM1k, mitigating bias evaluation caused by overfitting and data contamination.

**Correlation with Compression Efficiency.** Compression efficiency has been empirically proven that represent intelligence well (Du et al., 2024) even linearly (Huang et al., 2024a), well aligned with the belief that compression is closely connected to intelligence (Deletang et al., 2024). Following Huang et al. (2024a), we use BPC-Loss in Arxiv papers tagged with "Math" to measure compression efficiency as a surrogate. Figure 4 shows the correlation between BPC-Loss and GSM8K/MathCheck-GSM. The left sub-figure reveals that a single traditional benchmark like GSM8K cannot adequately reflect genuine mathematical ability, as indicated by the low Pearson correlation coefficient ($p = -0.822$). Many models, such as the Qwen series, deviate significantly from the regression line. In contrast, the right sub-figure displays the correlation with our MATHCHECK-GSM, demonstrating that MATHCHECK-GSM exhibits a significantly better correlation with genuine intelligence, with a Pearson correlation coefficient of $p = -0.915$. Our method shows that many models, such as the Qwen series, have scores on our benchmark that align more accurately with their true mathematical abilities. It shows that our design can represent mathematical intelligence more linearly.

## 4 BEHAVIOR ANALYSIS

MATHCHECK contains multi-dimensional information for evaluation, therefore we can observe the behaviors of the models on it to help analyze the models.

**Behavior of Math Models.** Recently, some works claim that math reasoning ability is greatly improved by training on massive amounts of math solving data. To validate whether their mathematical reasoning ability really improves, we examine the behaviors of the math models and their base models on MATHCHECK. As shown in Figure 5, compared with the base model, the performance of DeepSeek-Math-7B-Instruct/RL on solving units is greatly improved. However, the performance improvement on other units is limited, or even downward. The same phenomenon can be observed on MetaMath. It implies that training solely on massive solving data (Yue et al., 2023; Li et al., 2024a; Tang et al., 2024) is not the right direction to improve mathematical reasoning ability. Instead, training models with diverse mathematical data, beyond just solving, should be considered.

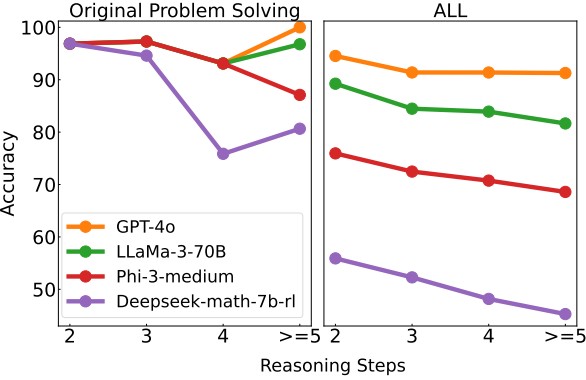
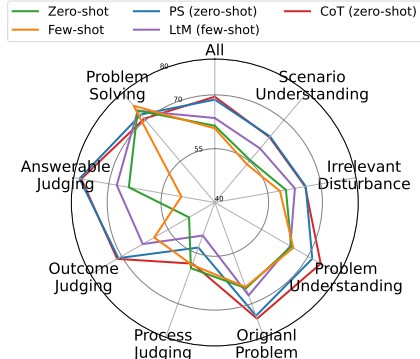

Figure 6: Performance on different complexity levels (i.e., reasoning steps) of MATHCHECK-GSM.

Figure 7: Different prompting technologies on MATHCHECK-GSM.

**Reasoning Consistency.** We analyze the reasoning consistency of generalist models across each unit in MATHCHECK, and the detailed results are shown in Appendix A and B. We can see most of them show good reasoning consistency since they achieve similar scores on each unit, such as GPT series, Llama-3 series and Mixtral series on MATHCHECK-GSM and GPT series on MATHCHECK-GEO. This is an interesting finding as it substantiates our assertion: *a model that really understands a problem can robustly work well on multiple related tasks.* Meanwhile, we also find that some models perform reasoning inconsistently. For example, Qwen1.5-72B-chat, Claude-3-Haiku and Phi-3-Medium show excellent performance on the solving task but much worse in other units of MATHCHECK-GSM. On MATHCHECK-GSM, Internet-VL achieves a high score of 40.0 on the original problem solving but decreases considerably when the problem switches to other robustness variants. These abnormal inconsistency behaviors of generalist models are highly similar to those mathematical models, revealing that they may conduct excessive decoration on original benchmarks.

**Behavior on Different Complexity Levels.** We categorize the complexity of problems based on the number of reasoning steps of the original problems, and select representative models of varying sizes for evaluation, as depicted in Figure 6. We can observe that the models' accuracy on the original problem solving fluctuates and does not show an obvious downward trend as the problems are more difficult. While the score "ALL" shows a steady downward trend, it implies that MATHCHECK better demonstrates the reasoning skills and capabilities required when problems become difficult.

**Behavior on Different Prompting Technologies.** We evaluate five prompting techniques including Zero-shot, Few-shot (Brown et al., 2020), CoT (Wei et al., 2022), Least to Most prompting (Zhou et al., 2022), and Plan-and-Solve prompting (Wang et al., 2023b). The results of GPT-3.5-Turbo on MATHCHECK-GSM are illustrated in Figure 7. Overall, Chain of Thought (CoT) and Plan-and-Solve (PS) in the zero-shot setting demonstrate superior performance, though this is not consistently the case across all tasks and settings. In contrast, the Few-shot prompt generally yields worse results than the Zero-shot prompt. Through detailed analyses, we find that the math reasoning generalization of LLMs is sensitive to Few-shot samples, which inspires us that Zero-shot with advanced prompt techniques (e.g., CoT or PS) may be a better choice in mathematical reasoning tasks.

## 5 MATHCHECK APPLIED TO OTHER REASONING TASKS

MATHCHECK can be adapted to other reasoning tasks beyond mathematical problems. We attempt the migration of the MATHCHECK paradigm in both commonsense reasoning and code generation.

**Commonsense Reasoning.** It requires LLMs to apply parametric knowledge to reason and solve problems. We choose the date understanding task in Big-bench (bench authors, 2023) as testbed since it is wildly used to measure commonsense reasoning ability (Wei et al., 2022). Appendix E.1 shows the case of applying MATHCHECK to date understanding. Similar to math reasoning, date understanding is a numerical reasoning task, therefore it can easily utilize variants of each unit in MATHCHECK. With MATHCHECK, raw data of date understanding have various test variants to examine the reasoning robustness and task generalization, helping us comprehensively evaluate models' date understanding.

**Code Generation.** We would like to show the possibility of transforming MATHCHECK in some real-world reasoning tasks such as code generation. Appendix E.2 demonstrates a case of applying MATHCHECK to code generation. Unlike numerical reasoning, the adaptation of code generation should consider task relevance. For real-world tasks such as agents and robotics application, multiple variants reflects the diversity of environment and user requirements.

# 6 RELATED WORK

**Benchmarks of Textual Mathematical Reasoning.** Numerous benchmarks have been proposed to evaluate the mathematical reasoning capabilities including (Amini et al., 2019; Cobbe et al., 2021; Frieder et al., 2024). Some datasets, such as the elementary-level GSM8k (Cobbe et al., 2021). Consequently, more challenging datasets have been introduced, including those at the high-school level (Hendrycks et al., 2021), university level (Sawada et al., 2023; Zheng et al., 2021) and olympic level (Huang et al., 2024b). Additionally, to provide a more comprehensive evaluation of mathematical reasoning abilities, numerous benchmarks have been developed that measure the robustness of mathematical reasoning (Li et al., 2024b), including semantic perturbations (Wang et al., 2023a; Zhou et al., 2024), reverse problem-solving (Yu et al., 2023; Berglund et al., 2023), irrelevant distractions (Shi et al., 2023; Li et al., 2023) and functional variation questions (Srivastava et al., 2024; Gulati et al., 2024). Above benchmarks paradigm can not comprehensively reflect reasoning ability at a given level. Therefore, MATHCHECK tries to go for better reasoning benchmark paradigm.

**Benchmarks of Visual Mathematical Reasoning.** Recently, multi-modal large language models have demonstrated outstanding capabilities in visual-language reasoning tasks (Allaway et al., 2022; Chen et al., 2023b; Yang et al., 2023; Team et al., 2023). Several benchmarks (Lin et al., 2014; Antol et al., 2015; Hudson & Manning, 2019; Marino et al., 2019; Mobasher et al., 2022) have been introduced to assess the visual reasoning capabilities of multi-modal large language models across various modalities including abstract scenes, geometric diagrams, graphics, and charts (Lu et al., 2021; Chen et al., 2021; 2022; Masry et al., 2022; Kazemi et al., 2023; Lu et al., 2023). MATHCHECK-GEO offers a comprehensive evaluation and testing platform for the research on visual math reasoning.

**Benchmarks of Reasoning Consistency.** Prior studies have identified limitations in reasoning consistency. Wu et al. (2023) designed counterfactual tasks to demonstrate that LLMs often rely on memorization to address general reasoning tasks. Berglund et al. (2023) found that LLMs struggle to answer inverse questions such as "B is A" after training on "A is B". In code reasoning, Gu et al. (2024) and Liu et al. (2024a) observed that LLMs successfully generate solution but fail to correct the wrong one. Similarly, Oh et al. (2024) found the gap between generation and evaluation in TriviaQA (Joshi et al., 2017). These findings inspire the design of MATHCHECK.

**Strategies of Improving Mathematical Reasoning.** Community has made significant efforts to enhance mathematical reasoning. In pre-training stage, previous works focus on collecting (Wang et al., 2023d; Paster et al., 2024; Shao et al., 2024) and synthesizing (Akter et al., 2024) math documents. In addition, Lin et al. (2024) selected key tokens in math data during pre-training. In post-training, numerous works generated massive problem-solving data for SFT (Yue et al., 2023; Li et al., 2024a; Tang et al., 2024). Besides, reinforcement learning such as GRPO (Shao et al., 2024) PRM (Lightman et al., 2024) can further improve reasoning ability. In inference, prompt and search strategies make LLMs reasoning better (Zhou et al., 2022; Wang et al., 2023b; Yao et al., 2024a).

# 7 CONCLUSION

In this paper, we argue that if a model really understands a problem, it should be able to successfully solve various tasks and variations of that problem. Based on this, we introduce **MATHCHECK**, a checklist for testing task generalization and reasoning robustness, along with an automatic tool for efficient checklist generation. MATHCHECK provides a clear view of model performance across dimensions, enabling more comprehensive evaluation. Using it, we develop **MATHCHECK-GSM** for textual reasoning and **MATHCHECK-GEO** for multi-modal reasoning. We evaluate massive (M)LLMs and conduct detailed analysis of model behaviors on MATHCHECK. Subsequently, we reveal that MATHCHECK better reflects reasoning abilities than prior benchmark paradigm. Finally, we show the potential of applying MATHCHECK paradigm to other reasoning tasks. We hope our practice can constitute a significant stride towards better reasoning benchmark paradigm.

ACKNOWLEDGEMENTS

The work was partially supported by the following: National Natural Science Foundation of China under No. 92370119, 62436009, 62276258 and 62376113, XJTLU Funding REF-22-01-002, the Science and Technology Development Fund of Macau SAR (Grant Nos. 0007/2024/AKP, FD-CT/0070/2022/AMJ, FDCT/060/2022/AFJ), the UM and UMDF (Grant Nos. MYRG-GRG2023-00006-FST-UMDF, MYRG-GRG2024-00165-FST-UMDF, EF2024-00185-FST, EF2023-00151-FST, EF2023-00090-FST), and the Commonwealth Cyber Initiative (CCI).

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

APPENDIX

## A  HEATMAP OF MATHCHECK-GSM

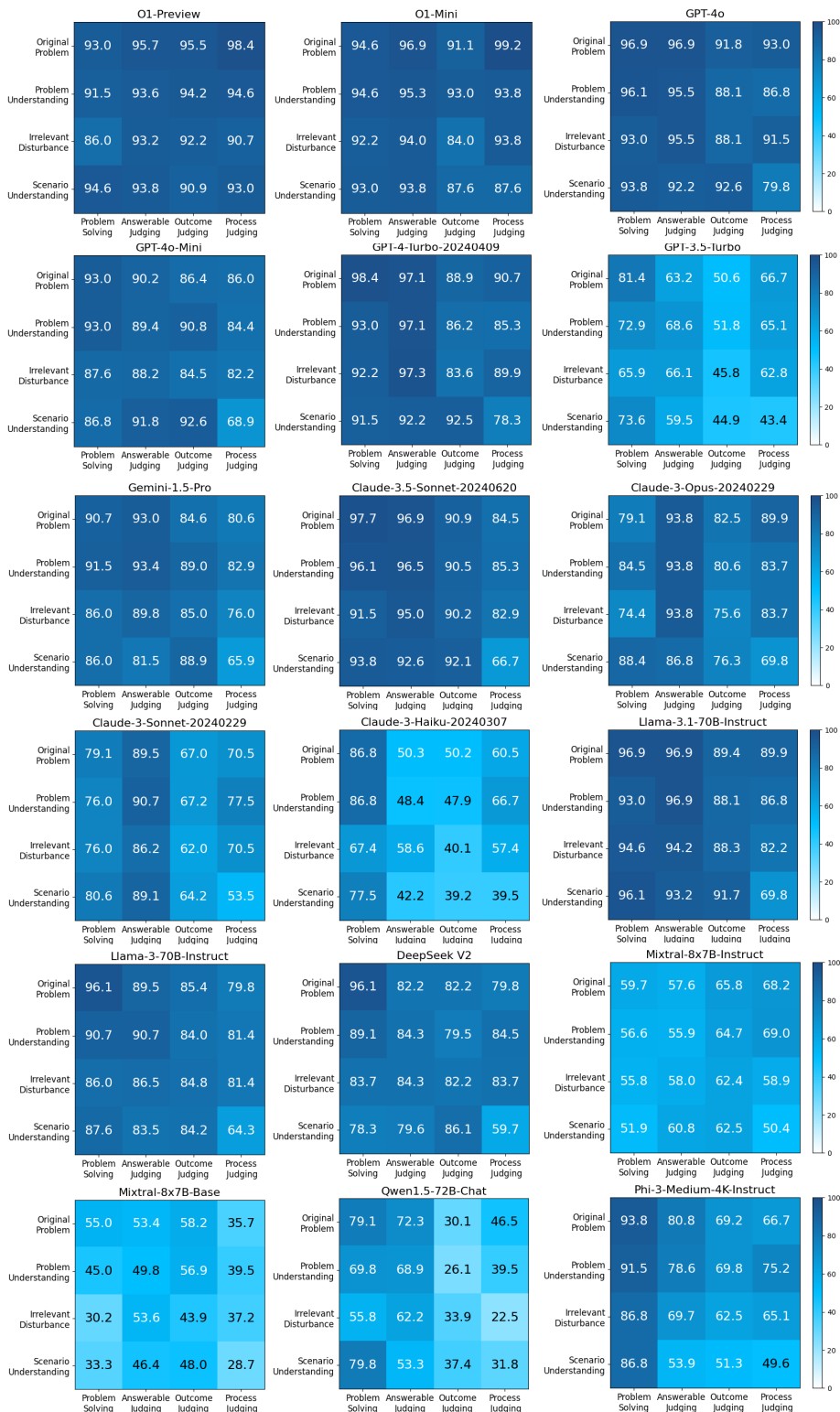

Figure 8: Visualized heatmap of MATHCHECK-GSM - Part 1.

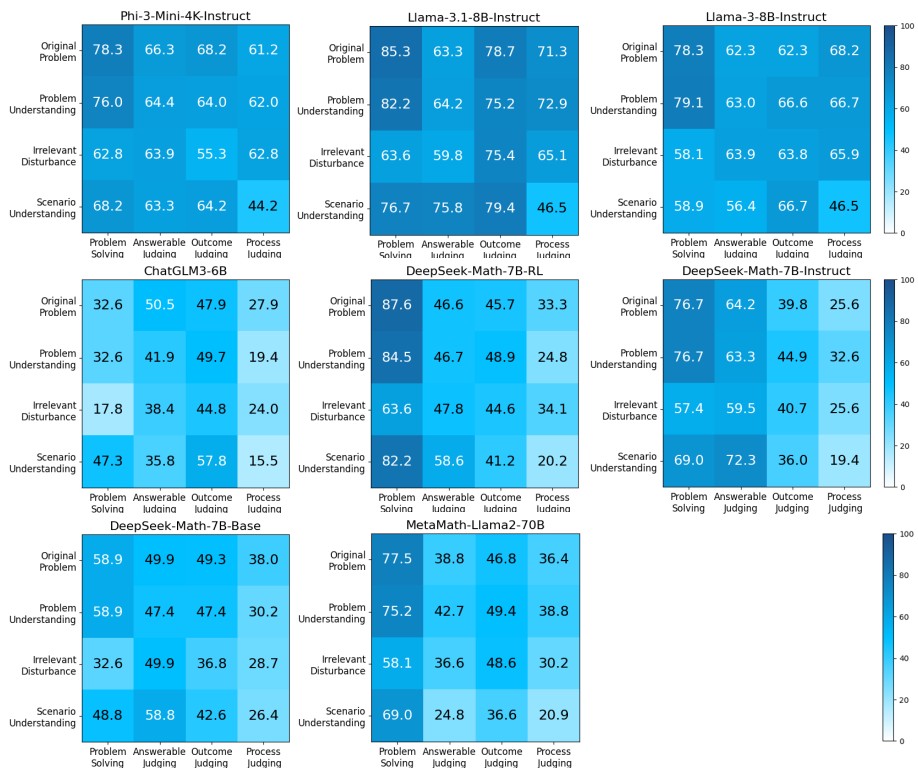

Figure 9: Visualized heatmap of MATHCHECK-GSM - Part 2.

# B  HEATMAP OF MATHCHECK-GEO

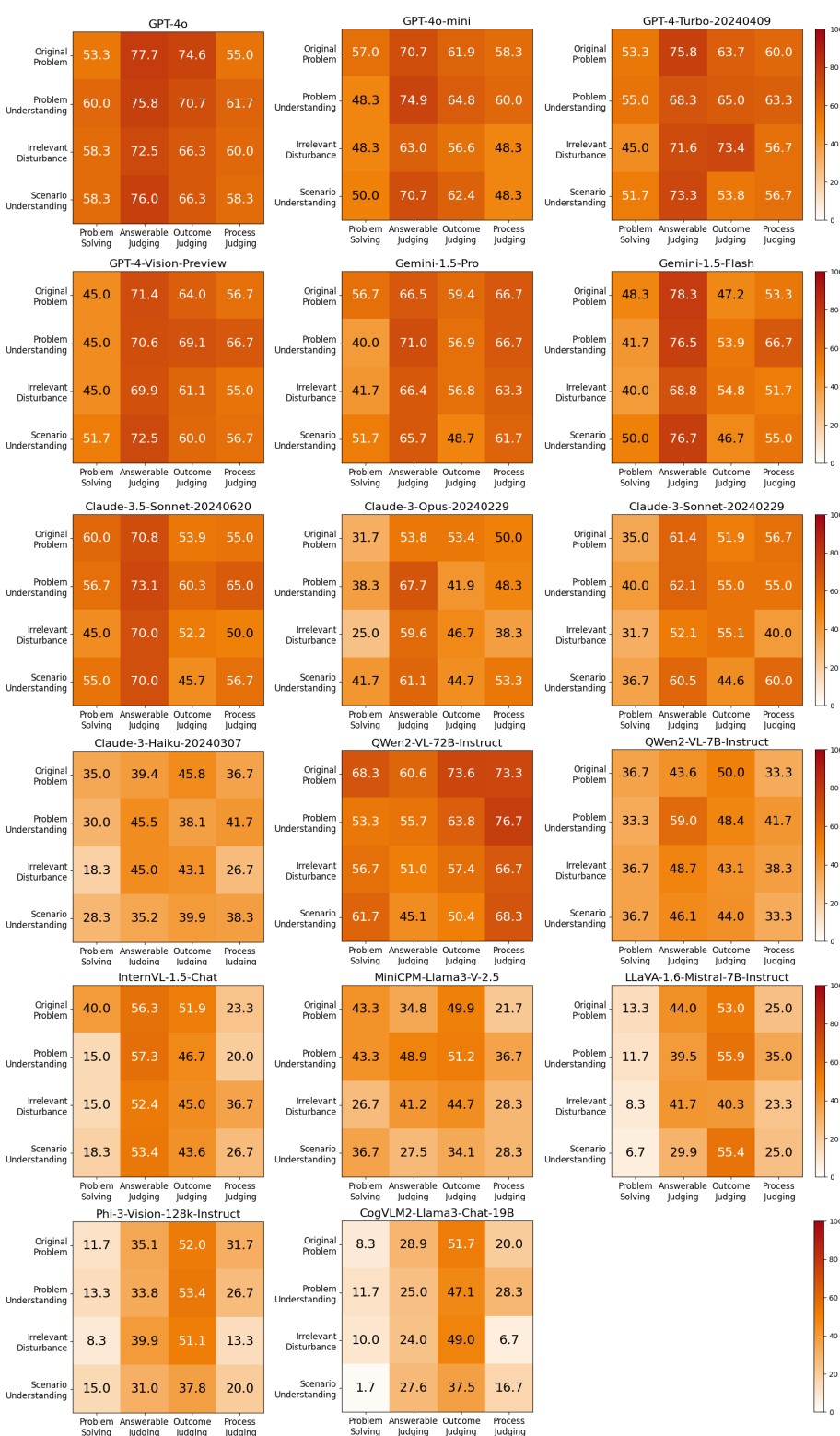

Figure 10: The visualized heatmap of MATHCHECK-GEO.

# C  DATA STATISTICS AND QUALITY

## C.1  OVERVIEW OF DATA

Table 3 and Table 4 show the data statistics of MATHCHECK-GSM and MATHCHECK-GEO. Table 5 shows the data statistics of each group in MATHCHECK-GSM and MATHCHECK-GEO. In each group, since answerable judging and outcome judging are binary-classification tasks, we try our best to include two different labels in these units for fair evaluation.

Table 3: Data statistics of MATHCHECK-GSM

|  | Problem Solving | Answerable Judging | Outcome Judging | Process Judging |
|---|---|---|---|---|
| **Original Problem** | 129 | 258 | 258 | 129 |
| **Problem Understanding** | 129 | 258 | 258 | 129 |
| **Irrelevant Disturbance** | 129 | 258 | 258 | 129 |
| **Scenario Understanding** | 129 | 258 | 258 | 129 |

Table 4: Data statistics of MATHCHECK-GEO

|  | Problem Solving | Answerable Judging | Outcome Judging | Process Judging |
|---|---|---|---|---|
| **Original Problem** | 60 | 120 | 120 | 60 |
| **Problem Understanding** | 60 | 120 | 120 | 60 |
| **Irrelevant Disturbance** | 60 | 120 | 120 | 60 |
| **Scenario Understanding** | 60 | 120 | 120 | 60 |

Table 5: Data statistics of each group in MATHCHECK-GSM and MATHCHECK-GEO

|  | Problem Solving | Answerable Judging | Outcome Judging | Process Judging |
|---|---|---|---|---|
| **Original Problem** | 1 | 2 | 2 | 1 |
| **Problem Understanding** | 1 | 2 | 2 | 1 |
| **Irrelevant Disturbance** | 1 | 2 | 2 | 1 |
| **Scenario Understanding** | 1 | 2 | 2 | 1 |

## C.2 Effectiveness of GPT-4-turbo Rewriting

In the process of human evaluation, we selected three graduate students as human annotators, all of them possess the mathematical skills required for evaluating the generated data. Our human evaluation principle is that the generated mathematical problems should maintain the correctness of mathematical logic. For example, in the "Problem Understanding", the generated question should not alter the logical structure of original question, which ensures the consistency between rewritten question and answer. The generated data will be marked as a failure if any of annotators determines that the generation failed. Furthermore, annotators corrected each failed data instead of discarding them. This approach ensures our dataset is entirely accurate and the evaluation results are reliable.

We conduct statistics on the pass rate of MATHCHECK-GSM rewritten by GPT4-turbo, as shown in Table 6. It can be seen that the rewriting pass rate is high, which reflects the effectiveness of our generation method. The success rate of Problem Understanding and Scenario Understanding is higher than 90%. There is a pass rate of 86.82% in the Irrelevant Disturbance and 81.40% in Wrong Step Rewriting. It provides references when we use MATHCHECK generation.

Table 6: Pass rate (%) checked by human annotators for the data generated by GPT4-turbo.

| Rewriting Type | Problem Understanding | Irrelevant Disturbance | Scenario Understanding | Unanswerable Question Rewriting | Wrong Step Rewriting |
|---|---|---|---|---|---|
| Human Pass Rate | 93.02 | 86.82 | 91.47 | 85.38 | 81.40 |

## C.3 Discussion of data bias generated by GPT

While we acknowledge there are possible self-bias in LLM-rewritten questions, we assert that this bias is acceptable and does not undermine the conclusions or rationality of MATHCHECK. This is supported by considerations across several dimensions.

**Motivations.** The motivation behind MATHCHECK is to establish a paradigm that mitigates benchmark hacking in the evaluation of mathematical reasoning, thereby revealing the genuine mathematical reasoning abilities of language models more comprehensively. Rewriting is an integral part of the MATHCHECK pipeline, which can naturally be performed by either humans or LLMs. While we acknowledge that involving experts in the rewriting process might be the fairest approach, the scalability of this method is a significant concern, as noted in several of today's LLM benchmarks, such as Arena Hard (Li et al., 2024c) and MT-Bench (Zheng et al., 2023), due to the high associated costs. To enhance scalability and practicality, we opted to use LLMs as the rewriters. Given that GPT-4 is widely recognized as the most advanced model accessible to the public, we believe that choosing GPT-4 as the rewriter is the closest approximation to the quality of expert human rewriting.

**Human-Checked Questions.** In fact, for the data construction which the LLM participates in, we mainly utilize the powerful rewriting ability of LLMs to edit the seed math problem instead of generating a new one from scratch. Moreover, we manually check the generated text to avoid some unnatural generated text.

**Experimental Results and Analysis.** On one hand, although the data are generated by GPT-4-Turbo in our experiments, they do not bring extra benefits to GPT-Family models to make them obviously outperform others. As shown in Table 1, the performance of Claude-3.5-sonnet is similar with GPT-4-Turbo, and even much better than GPT-4o-mini, which follows the commonsense on these LLMs. On the other hand, we compare the experimental results on Non-GPT-Rewritten and GPT-Rewritten Questions. In some data constructions where the LLM is not involved, GPT4-family exhibits the same performance ranking as the score "All". Specifically, the samples in Original Problem&Outcome Judging (**OP-OJ**) belong to Non-GPT-Rewiritten Questions, which are generated based on the rules. Table 7 shows that the performance ranking on non-LLM-generated data is close to the score "All", where GPT-series continues to perform better than other advanced models. All of these results verify that the possible bias to GPT models is acceptable in our MATHCHECK.

Table 7: Model performance on Non-GPT-Rewiritten Questions of MATHCHECK-GSM

| Models | All | OP-OJ |
|---|---|---|
| GPT-4o | 92.0 | **91.8** |
| GPT-4-Turbo-20240409 | 90.9 | **88.9** |
| Gemini-1.5-Pro | 86.3 | 84.6 |
| Claude-3-Opus-20240229 | 83.5 | 82.5 |
| Llama-3-70B-Instruct | 84.7 | 85.4 |

## D  EVALUATION SETUP

We conduct evaluations of multiple representative generalist and mathematical models on our MATH-CHECK benchmark. For MATHCHECK-GSM, the evaluation models encompass: (a) Generalist models, including proprietary models such as O1-Preview (OpenAI, 2024e), O1-Mini (OpenAI, 2024d), GPT-4o (OpenAI, 2024a), GPT-4o-mini (OpenAI, 2024b), GPT-4-Turbo (Achiam et al., 2023), GPT-3.5-Turbo (OpenAI, 2022), Gemini-1.5-Pro (Team et al., 2023), Claude-3 (Anthropic, 2024a), Claude-3.5-Sonnet Anthropic (2024b), Llama-3[2], Llama-3.1[3], DeepSeek V2 (Shao et al., 2024), Mixtral 8 x 7B (Jiang et al., 2024), Qwen1.5 (Bai et al., 2023), Phi-3 (Abdin et al., 2024), and ChatGLM3 (Du et al., 2022); (b) Mathematical models, including DeepSeek-Math (Shao et al., 2024) and MetaMath (Yu et al., 2023). For MATHCHECK-GEO, we conduct evaluations on generalist models: (a) proprietary models such as GPT-4o (OpenAI, 2024a), GPT-4o-mini (OpenAI, 2024b), GPT-4-Turbo (Achiam et al., 2023), GPT-4-vision (OpenAI, 2024c), Gemini-1.5-Pro (Team et al., 2023), Claude-3.5-Sonnet Anthropic (2024b) and Claude-3 (Anthropic, 2024a); (b) open-source models including Qwen2-VL (Wang et al., 2024), InternVL-1.5 (Chen et al., 2023c), Phi-3-Vision (Abdin et al., 2024), LLaVA-1.6-Mistral-7B-Instruct (Liu et al., 2024b), MiniCPM-Llama3-V-2.5 (Yao et al., 2024b) and CogVLM2-Llama3 (Wang et al., 2023c).

For Problem Solving and Process Judging tasks, we employ accuracy as the evaluation measure. For Outcome Judging and Answerable Judging tasks, we utilize Macro-F1 as the metric. We employ a zero-shot setting for generalist models and a few-shot setting (two-shot) for base models and mathematical models to enhance their ability to follow specific instructions and tasks. All the prompts used for evaluating (M)LLMs are provided in Appendix F.1.

For all the close-resourced models, we utilize the default hyper-parameters, setting the temperature to 0 and the max tokens to 1,024. Similarly, for all open-source models, the parameters are uniformly configured as follows: $do\_sample$ is set to False, $max\_gen\_len$ is set to 512, and the temperature is set to 0.1.

---

[2] https://ai.meta.com/blog/meta-llama-3
[3] https://ai.meta.com/blog/meta-llama-3-1

# E MATHCHECK APPLIED TO OTHER REASONING TASKS

Figure 11: Case of MATHCHECK in Date Understanding.

## E.1 DATE UNDERSTANDING

To show that our proposed benchmark paradigm MATHCHECK can be adapted to other reasoning tasks beyond mathematical problems, we try to transform some representative reasoning task into MATHCHECK paradigm. We firstly apply it in commonsense reasoning, which requires LLMs to apply world knowledge to reason and solve problems. Specifically, we choose the date understanding task in Big-bench (bench authors, 2023) since it is a wildly used task to measure commonsense reasoning ability (Wei et al., 2022).

Figure 11 shows the case of applying MATHCHECK to date understanding. Similar to mathematical reasoning, date understanding is a numerical reasoning task, therefore it can easily utilize variants of each unit in MATHCHECK. For example, in Irrelevant Disturbance, we can add some irrelevant date conditions to cause disturbance. In scenario understanding, we can ask for other variables in order to examine whether models have a comprehensive understanding of this date knowledge. This case demonstrates the high adaptability of MATHCHECK to commonsense reasoning task especially numerical reasoning.

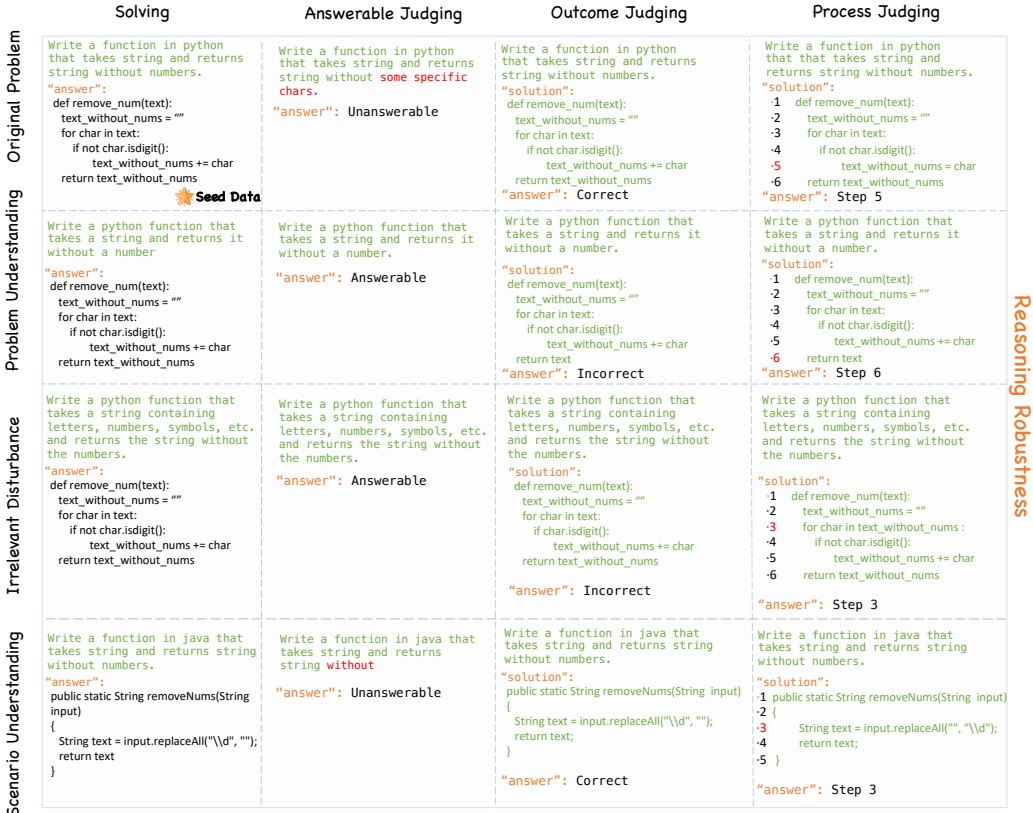

Figure 12: Case of MATHCHECK in Code Generation.

## E.2 CODE GENERATION

In addition to commonsense reasoning task, we would like to show the possibility of transforming MATHCHECK in some real-world reasoning tasks. Specifically, we choose the code generation task due to its high relevance to Text2Sql, agents and robotics. Figure 12 demonstrates a case of applying MATHCHECK to code generation. Unlike numerical reasoning tasks, the adaptation of code generation needs to consider task relevance. For example, in Scenario Understanding, we can ask models to write the same function in other program languages (Python to Java in our case) in order to examine whether models have a comprehensive understanding of this function requirements. It shows that MATHCHECK have potential for real-world tasks such as agents and robotics application. Meanwhile, we encourage researchers to design more specific variants towards their reasoning task on MATHCHECK framework to test reasoning robustness and task generalization.

# F  PROMPT LIST

## F.1  EVALUATION PROMPT

```
You are an AI assistant that determines whether math problems are solved
correctly. Answer the question. Finally give the answer in the format:
The answer is: ...

Question: [QUESTION]
Answer:
```

1: Zero-shot Prompt of Problem Solving

```
You are an AI assistant that determines whether math problems are solved
correctly. I will first give you a math problem and its solution, help me
 judge whether the final answer is correct or incorrect. Give your
judgment between Correct or Incorrect. Finally summarize your answer in
the format:
The answer is: ...

Question: [QUESTION]
Solution: [SOLUTION]
Judgement:
```

2: Zero-shot Prompt of Outcome Judging

```
You are an AI assistant that identify which step begins the error in
solution. I will give you a math problem along with a wrong solution.
Please help me identify the step where the errors begin. Finally give the
 wrong step in the format:
The answer is: Step i

Question: [QUESTION]
Solution: [SOLUTION]
Judgement:
```

3: Zero-shot Prompt of Process Judging

```
You are an AI assistant that determines whether math problems are
answerable or unanswerable. Please analyze whether the question provides
sufficient information to obtain an answer. Give your judgment between
Answerable or Unanswerable. Finally summarize your answer in the format:
The answer is: ...

Question: [QUESTION]
Judgement:
```

4: Zero-shot Prompt of Answerable Judging

```
You are an AI assistant to help me solve math problems. Answer the
question. Finally give the answer in the format: The answer is: ...
Follow the given examples and answer the question.

Question: Leah had 32 chocolates and her sister had 42. If they ate 35,
how many pieces do they have left in total?
Answer: Step 1: Originally, Leah had 32 chocolates.
Step 2: Her sister had 42. So in total they had 32 + 42 = 74.
Step 3: After eating 35, they had 74 - 35 = 39.
The answer is 39.

Question: Jason had 20 lollipops. He gave Denny some lollipops. Now Jason
 has 12 lollipops. How many lollipops did Jason give to Denny?
```

```
Answer: Step 1: Jason started with 20 lollipops.
Step 2: Then he had 12 after giving some to Denny.
Step 3: So he gave Denny 20 - 12 = 8.
The answer is 8.

Question: [QUESTION]
Answer:
```

5: Few-shot Prompt of Problem Solving

```
You are an AI assistant that determines whether math problems are solved
correctly. I will first give you a math problem and its solution, help me
 judge whether the final answer is correct or incorrect.

Give your judgment between Correct or Incorrect. Finally summarize your
answer in the format: The answer is: ...
Follow the given examples and give your judgment.

Question: Leah had 32 chocolates and her sister had 42. If they ate 35,
how many pieces do they have left in total?
Solution: Step 1: Originally, Leah had 32 chocolates.
Step 2: Her sister had 42. So in total they had 32 + 42 = 74.
Step 3: After eating 35, they had 74 - 35 = 39.
The answer is 39.
Judgment: Step 1 and Step 2 accurately calculate the total number of
chocolates they both had originally.
Step 3 correctly calculates how many they have left after eating 35
chocolates.
The answer is: Correct.

Question: Jason had 20 lollipops. He gave Denny some lollipops. Now Jason
 has 12 lollipops. How many lollipops did Jason give to Denny?
Solution: Step 1: Jason started with 20 lollipops.
Step2: Then he had 12 after giving some to Denny.
Step3: So he gave Denny 20 + 12 = 8.
The answer is 32.
Judgment: Jason ended up with 12 lollipops after giving some to Denny,
having started with 20. Therefore, the calculation to find out how many
lollipops Jason gave to Denny should be:20 - 12 = 8.
The answer is: Incorrect.

Question: [QUESTION]
Solution: [SOLUTION]
Judgement:
```

6: Few-shot Prompt of Outcome Judging

```
You are an AI assistant that identify which step begins the error in
solution. I will give you a math problem along with a wrong solution.
Please help me identify the step where the errors begin.

Finally give the wrong step in the format: The answer is: Step I
Follow the given examples and give your judgment.

Question: Leah had 32 chocolates and her sister had 42. If they ate 35,
how many pieces do they have left in total?
Solution: Step 1: Originally, Leah had 32 chocolates.
Step 2: Her sister had 42. So in total they had 32 + 42 = 84.
Step 3: After eating 35, they had 84 - 35 = 49.\nThe answer is 49.
Judgment: The judgment of the given steps is as follows:
Step 1: Correctly states Leah's initial amount of chocolates.
```

```
Step 2: Incorrectly calculates the total number of chocolates both Leah
and her sister had originally.
The answer is: Step 2.

Question: Jason had 20 lollipops. He gave Denny some lollipops. Now Jason
 has 12 lollipops. How many lollipops did Jason give to Denny?
Solution: Step 1: Jason started with 20 lollipops.
Step 2: Then he had 12 after giving some to Denny.
Step 3: So he gave Denny 20 + 12 = 8.
The answer is 32.
Judgment: The correct method to find out how many lollipops Jason gave to
 Denny would be to subtract the amount he had left from the amount he
started with: 20 - 12 = 8. Thus, The reasoning error begins at Step 3.
The answer is: Step 3.

Question: [QUESTION]
Solution: [SOLUTION]
Judgement:
```

7: Few-shot Prompt of Process Judging

```
You are an AI assistant that determines whether math problems are
answerable or unanswerable. Please analyze whether the question provides
sufficient information to obtain an answer.

Give your judgment between Answerable or Unanswerable. Finally summarize
your answer in the format: The answer is: ...
Follow the given examples and give your judgment.

Question: Leah had 32 chocolates and her sister had 42. If they ate 35,
how many pieces do they have left in total?
Judgment: The question provides all necessary information to perform the
calculation.
The answer is: Answerable.

Question: Jason had 20 lollipops. He gave Denny some lollipops. How many
lollipops did Jason give to Denny?
Judgment: The question is not answerable as given. The reason is that
there is insufficient information to determine the exact number of
lollipops Jason gave to Denny.
The answer is: Unanswerable.

Question: [QUESTION]
Judgement:
```

8: Few-shot Prompt of Answerable Judging

## F.2 Data Generation Prompt

```
Your objective is to rewrite a given math question using the following
perturbation strategy. The rewritten question should be reasonable,
understandable, and able to be responded to by humans.

Perturbation strategy: Problem Understanding: It refers to transforming
the original problem into a new problem that uses different wording or
different sentence structures but does not change the solution of the
original problem.

The given question: {QUESTION}
Answer of the given question: {ANSWER}
```

```
Please rewrite the question using the specified perturbation strategy
while minimizing edits to avoid significant deviation in the question
content.
It is important to ensure that the rewritten question has only one
required numerical answer. You just need to print the rewritten question
without answer.
The rewritten question:
Question: {QUESTION}
Answer: {ANSWER}
Given step: {STEP}
The rewritten answer:
```

9: Prompt of Problem Understanding Rewriting

```
Your objective is to rewrite a given math question using the following
perturbation strategy. The rewritten question should be reasonable,
understandable, and able to be responded to by humans.

Perturbation strategy: Irrelevant Disturbance: It involves introducing
distracting conditions that have no impact on the final answer. These
introduced conditions should be relevant to the topic of the original
question and preferably include numerical values. However, the rewritten
problem must maintain an identical solution to that of the original
problem.

The given question: {QUESTION}
Answer of the given question: {ANSWER}

Please rewrite the question using the specified perturbation strategy
while minimizing edits to avoid significant deviation in the question
content.
It is important to ensure that the rewritten question has only one
required numerical answer. You just need to print the rewritten question
without answer.
The rewritten question:
Question: {QUESTION}
Answer: {ANSWER}
Given step: {STEP}
The rewritten answer:
```

10: Prompt of Irrelevant Disturbance Rewriting

```
Your objective is to rewrite a given math question using the following
perturbation strategy. The rewritten question should be reasonable,
understandable, and able to be responded to by humans.

Perturbation strategy: Unanswerable question: It refers to eliminating a
condition from the original question that is crucial for solving it while
 keeping the rest of the content unchanged. The rewritten problem should
no longer have a valid answer, as it lacks the constraint that was
removed.

The given question: {QUESTION}
Answer of the given question: {ANSWER}

Please rewrite the question using the specified perturbation strategy
while minimizing edits to avoid significant deviation in the question
content.
It is important to ensure that the rewritten question has only one
required numerical answer. You just need to print the rewritten question
without answer.
The rewritten question:
Question: {QUESTION}
Answer: {ANSWER}
```

```
Given step: {STEP}
The rewritten answer:
```

11: Prompt of Unanswerable Question Rewriting

```
You are an AI assistant to help me rewrite question into a declarative
statement when its answer is provided.
Follow the given examples and rewrite the question.

Question: How many cars are in the parking lot? The answer is 5.
Result: There are 5 cars in the parking lot.

Question: How many trees did the grove workers plant today? The answer is
 6.
Result: The grove workers planted 6 trees today.

Question: If they ate 35, how many pieces do they have left in total? The
 answer is 39.
Result: They have 39 pieces left in total if they ate 35.

Question: How many lollipops did Jason give to Denny? The answer is 8.
Result: Jason gave 8 lollipops to Denny.

Question: How many toys does he have now? The answer is 9.
Result: He now has 9 toys.

Question: How many computers are now in the server room? The answer is
29.
Result: There are 29 computers now in the server room.

Question: How many golf balls did he have at the end of wednesday? The
answer is 33.
Result: He had 33 golf balls at the end of Wednesday.

Question: How much money does she have left? The answer is 8.
Result: She has 8 money left.

Question: {QUESTION} The answer is {ANSWER}.
Result:
```

12: Prompt to Rewrite Question and Answer into a Declarative Statement

```
Following is a question and its correct solution. Rewrite the solution
according to following requirements: (1) Do not change the format (2)
Keep those steps before the given step unchanged (3) Make minor changes
to the given step so that the reasoning of this step and subsequent steps
 are incorrect, resulting in an incorrect answer.

Question: {QUESTION}
Answer: {ANSWER}
Given step: {STEP}
The rewritten answer:
```

13: Prompt to Generate the Wrong Step

# G CASE PROBLEMS

## G.1 CASE PROBLEMS IN MATHCHECK-GSM. PROBLEM GROUP ID: GSM-54

```
[Question]: Mike plays ping pong for 40 minutes. In the first 20 minutes,
 he scores 4 points. In the second 20 minutes, he scores 25% more points.
 How many total points did he score?
[Answer]: 9.0
```

14: Problem Solving - Original Problem

```
[Question]: During a 40-minute ping pong session, Mike scores 4 points in
 the initial half. In the latter half, he manages to increase his score
by 25% compared to the first half. What is the total score Mike achieved
in this session?
[Answer]: 9.0
```

15: Problem Solving - Problem Understanding

```
[Question]: Mike plays ping pong in a local tournament and decides to
practice for 40 minutes before the first match. During his practice
session, in the first 20 minutes, while intermittently checking his phone
 and hydrating, he manages to score 4 points. In the following 20 minutes
, feeling more warmed up and despite a short break to adjust his paddle's
 grip tape, he scores 25% more points than in the first session.
Considering these distractions, how many total points did Mike score in
his 40-minute practice session?
[Answer]: 9.0
```

16: Problem Solving - Irrelevant Disturbance

```
[Question]: Mike plays ping pong for 40 minutes.  In the first 20 minutes
, he scores x points.  In the second 20 minutes, he scores 25% more
points. He scored 9 total points. What is the value of unknown variable x
?
[Answer]: 4.0
```

17: Problem Solving - Scenario Understanding

```
[Question]: Mike plays ping pong for 40 minutes. In the first 20 minutes,
 he scores 4 points. In the second 20 minutes, he scores 25% more points.
 How many total points did he score?
[Answer]: Answerable
```

18: Answerable Judging (*Answerable*) - Original Problem

```
[Question]: Mike plays ping pong for minutes. In the first 20 minutes, he
 scores 4 points. In the second 20 minutes, his performance increases by
25%. How many total points did he score?
[Answer]: Unanswerable
```

19: Answerable Judging (*Unanswerable*) - Original Problem

```
[Question]: During a 40-minute ping pong session, Mike scores 4 points in
 the initial half. In the latter half, he manages to increase his score
by 25% compared to the first half. What is the total score Mike achieved
in this session?
[Answer]: Answerable
```

20: Answerable Judging (*Answerable*) - Problem Understanding

```
[Question]: During a 40-minute ping pong session, Mike scores points in
the initial half. In the latter half, he manages to increase his score by
 25% compared to the first half. What is the total score Mike achieved in
 this session?
[Answer]: Unanswerable
```

21: Answerable Judging (*Unanswerable*) - Problem Understanding

```
[Question]: Mike plays ping pong in a local tournament and decides to
practice for 40 minutes before the first match. During his practice
session, in the first 20 minutes, while intermittently checking his phone
 and hydrating, he manages to score 4 points. In the following 20 minutes
, feeling more warmed up and despite a short break to adjust his paddle's
 grip tape, he scores 25% more points than in the first session.
Considering these distractions, how many total points did Mike score in
his 40-minute practice session?
[Answer]: Answerable
```

22: Answerable Judging (*Answerable*) - Irrelevant Disturbance

```
[Question]: Mike plays ping pong in a local tournament and decides to
practice for 40 minutes before the first match. During his practice
session, in the first 20 minutes, while intermittently checking his phone
 and hydrating, he manages to score points. In the following 20 minutes,
feeling more warmed up and despite a short break to adjust his paddle's
grip tape, he scores 25% more points than in the first session.
Considering these distractions, how many total points did Mike score in
his 40-minute practice session?
[Answer]: Unanswerable
```

23: Answerable Judging (*Unanswerable*) - Irrelevant Disturbance

```
[Question]: Mike plays ping pong for 40 minutes.  In the first 20 minutes
, he scores x points.  In the second 20 minutes, he scores 25% more
points. He scored 9 total points. What is the value of unknown variable x
?
[Answer]: Answerable
```

24: Answerable Judging (*Answerable*) - Scenario Understanding

```
[Question]: Mike plays ping pong for 40 minutes. In the first 20 minutes,
 he scores x points. In the second 20 minutes, he scores 25% more points.
 What is the value of unknown variable x?
[Answer]: Unanswerable
```

25: Answerable Judging (*Unanswerable*) - Scenario Understanding

```
[Question]: Mike plays ping pong for 40 minutes. In the first 20 minutes,
 he scores 4 points. In the second 20 minutes, he scores 25% more points.
 How many total points did he score?
[Solution]:
Step 1: Find the number of points Mike scored in the second 20 minutes.
To do this, take the number of points he scored in the first 20 minutes,
which is 4 points, and increase this by 25%.
Step 2: To calculate 25% of 4, divide 4 by 4 which gives 1 (since 25% is
the same as 1/4th).
Step 3: Add this 25% (which is 1 point) to the original 4 points to find
out how many points he scored in the second 20 minutes. So, 4 points + 1
point = 5 points.
Step 4: Now, to find the total number of points Mike scored in the 40
minutes, add the points scored in the first 20 minutes to the points
scored in the second 20 minutes.
```

```
So, 4 points (first 20 minutes) + 5 points (second 20 minutes) = 9 points
.
The answer is 9 points.
[Answer]: Correct
```

26: Outcome Judging (*Correct*) - Original Problem

```
[Question]: Mike plays ping pong for 40 minutes.  In the first 20 minutes
, he scores 4 points. In the second 20 minutes, he scores 25% more points
. How many total points did he score?
[Solution]:
Step 1: Mike scores 4 points in the first 20 minutes.
Step 2: Mike scores 25% more points in the second 20 minutes.
Step 3: Mike's total points are the sum of his points in the first and
second 20 minutes.
The answer is 28 points.
[Answer]: Incorrect
```

27: Outcome Judging (*Incorrect*) - Original Problem

```
[Question]: During a 40-minute ping pong session, Mike scores 4 points in
 the initial half. In the latter half, he manages to increase his score
by 25% compared to the first half. What is the total score Mike achieved
in this session?
[Solution]:
Step 1: Calculate Mike's score in the first half, which is given as 4
points.
Step 2: Calculate the 25% increase in score for the second half. \\(25\\%
 \\text{ of } 4 = \\frac{25}{100} \\times 4 = 1\\).
Step 3: Add this increase to the initial score to find the score for the
second half. \\(4 + 1 = 5\\).
Step 4: Combine the scores from both halves to get the total score. \\(4
\\text{ (first half) } + 5 \\text{ (second half) } = 9\\).
The answer is 9.
[Answer]: Correct
```

28: Outcome Judging (*Correct*) - Problem Understanding

```
[Question]: During a 40-minute ping pong session, Mike scores 4 points in
 the initial half. In the latter half, he manages to increase his score
by 25% compared to the first half. What is the total score Mike achieved
in this session?
[Solution]:
Step 1: Mike scores 4 points in the initial half, so his score after 20
minutes is 20 minutes \\* 4 points/20 minutes = 8 points.
Step 2: In the latter half, Mike's score increases by 25% compared to the
 first half, so his score after 20 minutes is 8 points + (8 points \\*
25% increase) = 8 points + 2 points = 10 points.
Step 3: Therefore, Mike's total score after 40 minutes is 10 points + 4
points (from the initial half) = 14 points.
The answer is 14 points.
[Answer]: Incorrect
```

29: Outcome Judging (*Incorrect*) - Problem Understanding

```
[Question]: Mike plays ping pong in a local tournament and decides to
practice for 40 minutes before the first match. During his practice
session, in the first 20 minutes, while intermittently checking his phone
 and hydrating, he manages to score 4 points. In the following 20 minutes
, feeling more warmed up and despite a short break to adjust his paddle's
 grip tape, he scores 25% more points than in the first session.
Considering these distractions, how many total points did Mike score in
his 40-minute practice session?
```

```
[Solution]:
Step 1: Identify the points scored by Mike in the first 20 minutes, which
 is 4 points.
Step 2: Calculate the number of points scored in the next 20 minutes by
finding 25% more than the points scored in the first session. 25% of 4
points is 1 (which is 0.25 * 4).
Step 3: Add the additional points to the first session's points to find
the total points scored in the second session. So, 4 points + 1 point = 5
 points in the second session.
Step 4: To find the total points scored in the 40-minute practice session
, add the points from the first and second sessions together.
So, 4 points + 5 points = 9 points.\n\nThe answer is 9 points.
[Answer]: Correct
```

30: Outcome Judging (*Correct*) - Irrelevant Disturbance

```
[Question]: Mike plays ping pong in a local tournament and decides to
practice for 40 minutes before the first match. During his practice
session, in the first 20 minutes, while intermittently checking his phone
 and hydrating, he manages to score 4 points. In the following 20 minutes
, feeling more warmed up and despite a short break to adjust his paddle's
 grip tape, he scores 25% more points than in the first session.
Considering these distractions, how many total points did Mike score in
his 40-minute practice session?
[Solution]:
Step 1: Calculate the points scored in the first 20 minutes.\n4 points in
 20 minutes = 2 points per minute
Step 2: Calculate the points scored in the second 20 minutes.\n25% more
points than in the first session = 1.25 x 2 points/minute = 2.5 points/
minute
Step 3: Calculate the total points scored in the 40-minute practice
session.\n2 points/minute x 20 minutes = 40 points\n2.5 points/minute x
20 minutes = 50 points
Step 4: Add the points scored in the first and second sessions.\n40
points + 50 points = 90 points
The answer is 90 points.
[Answer]: Incorrect
```

31: Outcome Judging (*Incorrect*) - Irrelevant Disturbance

```
[Question]: Mike plays ping pong for 40 minutes. In the first 20 minutes,
 he scores x points. In the second 20 minutes, he scores 25% more points.
 He scored 9 total points. What is the value of unknown variable x?
[Solution]:
Step 1: Define the given information: Mike plays ping pong for 40 minutes
, scores x points in the first 20 minutes, and scores 25% more points in
the second 20 minutes.
Step 2: Since he scores 25% more points in the second 20 minutes, we can
express the points scored in the second 20 minutes as \\(x + 0.25x\\).
Step 3: This can be simplified as \\(1.25x\\), representing the points
scored in the second 20 minutes.
Step 4: Since he scored 9 points in total, we can write the equation:\n
\\[x + 1.25x = 9\\]
Step 5: By simplifying the equation, we combine like terms:\n\\[2.25x =
9\\]
Step 6: To find the value of x, we divide both sides of the equation by
2.25:\n\\[x = \\frac{9}{2.25}\\]
Step 7: Simplify the division:\n\\[x = 4\\]
The answer is x = 4.
[Answer]: Correct
```

32: Outcome Judging (*Correct*) - Scenario Understanding

```
[Question]: Mike plays ping pong for 40 minutes. In the first 20 minutes,
 he scores x points. In the second 20 minutes, he scores 25% more points.
 He scored 9 total points. What is the value of unknown variable x?
[Solution]:
Step 1: Let's first find out how many points Mike scored in the first 20
minutes.
Step 2: Since Mike scored x points in the first 20 minutes, we know that
x/2 = 9 (since he scored 9 points in total).
Step 3: Now, we know that x = 2(9) = 18.
The answer is x = 18.
[Answer]: Incorrect
```

33: Outcome Judging (*Incorrect*) - Scenario Understanding

```
[Question]: Mike plays ping pong for 40 minutes. In the first 20 minutes,
 he scores 4 points. In the second 20 minutes, he scores 25% more points.
 How many total points did he score?
[Solution]:
Step 1: Find the number of points Mike scored in the second 20 minutes.
To do this, take the number of points he scored in the first 20 minutes,
which is 4 points, and increase this by 25%.
Step 2: To calculate 25% of 4, divide 4 by 4 which gives 1 (since 25% is
the same as 1/4th).
Step 3: Add this 25% (which is 2 points) to the original 4 points to find
 out how many points he scored in the second 20 minutes. So, 4 points + 2
 points = 6 points.
Step 4: Now, to find the total number of points Mike scored in the 40
minutes, add the points scored in the first 20 minutes to the points
scored in the second 20 minutes. So, 4 points (first 20 minutes) + 6
points (second 20 minutes) = 10 points.
The answer is 10 points.
[Answer]: Step 3
```

34: Process Judging - Original Problem

```
[Question]: During a 40-minute ping pong session, Mike scores 4 points in
 the initial half. In the latter half, he manages to increase his score
by 25% compared to the first half. What is the total score Mike achieved
in this session?
[Solution]:
Step 1: Calculate Mike's score in the first half, which is given as 4
points.
Step 2: Calculate the 25% increase in score for the second half. \\(4 \\
div 25 = 0.16\\).
Step 3: Add this increase to the initial score to find the score for the
second half. \\(4 + 0.16 = 4.16\\).
Step 4: Combine the scores from both halves to get the total score. \\(4
\\text{ (first half) } + 4.16 \\text{ (second half) } = 8.16\\).
The answer is 8.16.
[Answer]: Step 2
```

35: Process Judging - Problem Understanding

```
[Question]: Mike plays ping pong in a local tournament and decides to
practice for 40 minutes before the first match. During his practice
session, in the first 20 minutes, while intermittently checking his phone
 and hydrating, he manages to score 4 points. In the following 20 minutes
, feeling more warmed up and despite a short break to adjust his paddle's
 grip tape, he scores 25% more points than in the first session.
Considering these distractions, how many total points did Mike score in
his 40-minute practice session?
[Solution]:
```

```
Step 1: Identify the points scored by Mike in the first 20 minutes, which
 is 4 points.
Step 2: Calculate the number of points scored in the next 20 minutes by
finding 25% more than the points scored in the first session. 25% of 4
points is 1 (which is 0.25 * 4).
Step 3: Add the additional points to the first session's points to find
the total points scored in the second session. So, 4 points + 1 point = 6
 points in the second session.
Step 4: To find the total points scored in the 40-minute practice session
, add the points from the first and second sessions together. So, 4
points + 6 points = 10 points.
The answer is 10 points.
[Answer]: Step 3
```

36: Process Judging - Irrelevant Disturbance

```
[Question]: Mike plays ping pong for 40 minutes. In the first 20 minutes,
 he scores x points. In the second 20 minutes, he scores 25% more points.
 He scored 9 total points. What is the value of unknown variable x?
[Solution]:
Step 1: Define the given information: Mike plays ping pong for 40 minutes
, scores x points in the first 20 minutes, and scores 25% more points in
the second 20 minutes.
Step 2: Since he scores 25% more points in the second 20 minutes, we can
express the points scored in the second 20 minutes as \\(x + 0.25x\\).
Step 3: This can be simplified as \\(1.25x\\), representing the points
scored in the second 20 minutes.
Step 4: Since he scored 9 points in total, we can write the equation:\n
\\[x + 1.25x = 9\\]
Step 5: By simplifying the equation, we combine like terms:\n\\[2.25x =
9\\]
Step 6: To find the value of x, we divide both sides of the equation by
2.25:\n\\[x = \\frac{9}{2.25}\\]
Step 7: Simplify the division:\n\\[x = 5\\]\n\nThe answer is x = 5.
[Answer]: Step 7
```

37: Process Judging - Scenario Understanding

## G.2 CASE PROBLEMS IN MATHCHECK-GEO. PROBLEM GROUP ID: GEO-15

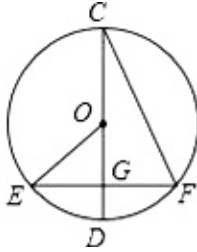

Figure 13: Geometry diagram for geometry problems in group 15.

```
[Question]: As shown in the figure, the diameter CD of \\odot O crosses
the midpoint G of chord EF, \\angle DCF = 20.0, then \\angle EOD is equal
 to ()\\degree
[Answer]: 40.0
```

38: Problem Solving - Original Problem

```
[Question]: In the circle with center O, diameter CD intersects the
midpoint G of the chord EF, and the measure of angle DCF is 20 degrees.
Determine the measurement of angle EOD in degrees.
[Answer]: 40.0
```

39: Problem Solving - Problem Understanding

```
[Question]: In the figure of circle O, the diameter CD intersects the
midpoint G of the chord EF. The length of the chord EF is 7.5 cm, which
is irrelevant to our angle measurements. The angle \\angle DCF is given
to be 20.0 degrees. We need to calculate the angle \\angle EOD. What is
the measure of this angle in degrees?
[Answer]: 40.0
```

40: Problem Solving - Irrelevant Disturbance

```
[Question]: As shown in the figure, the diameter CD of \\odot O crosses
the midpoint G of chord EF, \\angle DCF = x , \\angle EOD is equal to
40\\degree. What is the value of unknown variable x?
[Answer]: 20.0
```

41: Problem Solving - Scenario Understanding

```
[Question]: As shown in the figure, the diameter CD of \\odot O crosses
the midpoint G of chord EF, \\angle DCF = 20.0, then \\angle EOD is equal
 to ()\\degree
[Answer]: Answerable
```

42: Answerable Judging (*Answerable*) - Original Problem

```
[Question]: As shown in the figure, the diameter CD of \\odot O crosses
chord EF, \\angle DCF = 20.0, then \\angle EOD is equal to ()\\degree
[Answer]: Unanswerable
```

43: Answerable Judging (*Unanswerable*) - Original Problem

```
[Question]: In the circle with center O, diameter CD intersects the
midpoint G of the chord EF, and the measure of angle DCF is 20 degrees.
Determine the measurement of angle EOD in degrees.
[Answer]: Answerable
```

44: Answerable Judging (*Answerable*) - Problem Understanding

```
[Question]: In the circle with center O, diameter CD intersects the
midpoint G of the chord EF. Determine the measurement of angle EOD in
degrees.
[Answer]: Unanswerable
```

45: Answerable Judging (*Unanswerable*) - Problem Understanding

```
[Question]: In the figure of circle O, the diameter CD intersects the
midpoint G of the chord EF. The length of the chord EF is 7.5 cm, which
is irrelevant to our angle measurements. The angle \\angle DCF is given
to be 20.0 degrees. We need to calculate the angle \\angle EOD. What is
the measure of this angle in degrees?
[Answer]: Answerable
```

46: Answerable Judging (*Answerable*) - Irrelevant Disturbance

```
[Question]: In the figure of circle O, the diameter CD intersects the
midpoint G of the chord EF. The length of the chord EF is 7.5 cm, which
is irrelevant to our angle measurements. We need to calculate the angle
\\angle EOD. What is the measure of this angle in degrees?
[Answer]: Unanswerable
```

47: Answerable Judging (*Unanswerable*) - Irrelevant Disturbance

```
[Question]: As shown in the figure, the diameter CD of \\odot O crosses
the midpoint G of chord EF, \\angle DCF = x , \\angle EOD is equal to
40\\degree. What is the value of unknown variable x?
[Answer]: Answerable
```

48: Answerable Judging (*Answerable*) - Scenario Understanding

```
[Question]: As shown in the figure, the diameter CD of \\odot O crosses
the midpoint G of chord EF, \\angle DCF = x. What is the value of unknown
 variable x?
[Answer]: Unanswerable
```

49: Answerable Judging (*Unanswerable*) - Scenario Understanding

```
[Question]: As shown in the figure, the diameter CD of \\odot O crosses
the midpoint G of chord EF, \\angle DCF = 20.0, then \\angle EOD is equal
 to ()\\degree
[Solution]:
Step 1: Identify that the angle DCF is an angle formed by a chord (CF)
and a tangent (CD) at the point C.
Step 2: Recognize that the angle between a chord and a tangent at the
point of contact is equal to the angle in the alternate segment, which is
 angle ECF in this case.
Step 3: Thus, the measure of angle ECF is also 20 degrees.
Step 4: Observe that the diameter CD passes through the center O of the
circle and cuts the chord EF at its midpoint G.
Step 5: Recall that a line drawn from the center of the circle to the
midpoint of a chord is perpendicular to the chord.
Step 6: Therefore, triangle EOG is a right-angled triangle at G.
Step 7: Understand that angle EOD is double the angle ECF because angle
EOD is the angle at the center whereas angle ECF is at the circumference
based on the Circle Theorem.
Step 8: Calculate angle EOD as 2 times angle ECF, which is 2 * 20 degrees
.
Step 9: Perform the multiplication: 2 * 20 degrees = 40 degrees.
The answer is: Angle EOD measures 40 degrees.
[Answer]: Correct
```

50: Outcome Judging (*Correct*) - Original Problem

```
[Question]: As shown in the figure, the diameter CD of \\odot O crosses
the midpoint G of chord EF, \\angle DCF = 20.0, then \\angle EOD is equal
 to ()\\degree
[Solution]:
Step 1: Notice that chord EF is bisected by diameter CD at point G.
Step 2: Recognize that \\angle DCF is an angle formed by the diameter CD
and chord EF, so it is inscribed in the semicircle of \\odot O.
Step 3: Recall that an angle inscribed in a semicircle is a right angle;
hence \\angle DCF = 90\\degree (not the given 20\\degree, which seems to
be a mistake).
Step 4: Realize that \\angle EOD is the angle at the center of the circle
, and it stands on the arc EF.
Step 5: Understand that the angle at the center of a circle is twice the
angle at the circumference when both stand on the same arc.
Step 6: Since \\angle DCF is a right angle (90\\degree) and \\angle EOD
is twice that angle at the center, \\angle EOD is 2 * 90\\degree.
Step 7: Calculate 2 * 90\\degree to get 180\\degree.
The answer is \\angle EOD is equal to 180\\degree.
[Answer]: Incorrect
```

51: Outcome Judging (*Incorrect*) - Original Problem

```
[Question]: In the circle with center O, diameter CD intersects the
midpoint G of the chord EF, and the measure of angle DCF is 20 degrees.
Determine the measurement of angle EOD in degrees.
[Solution]:
Step 1: Identify that the angle DCF is an angle formed by a chord (CF)
and a tangent (CD) at the point C.
Step 2: Recognize that the angle between a chord and a tangent at the
point of contact is equal to the angle in the alternate segment, which is
 angle ECF in this case.
Step 3: Thus, the measure of angle ECF is also 20 degrees.
Step 4: Observe that the diameter CD passes through the center O of the
circle and cuts the chord EF at its midpoint G.
Step 5: Recall that a line drawn from the center of the circle to the
midpoint of a chord is perpendicular to the chord.
Step 6: Therefore, triangle EOG is a right-angled triangle at G.
Step 7: Understand that angle EOD is double the angle ECF because angle
EOD is the angle at the center whereas angle ECF is at the circumference
based on the Circle Theorem.
Step 8: Calculate angle EOD as 2 times angle ECF, which is 2 * 20 degrees
.
Step 9: Perform the multiplication: 2 * 20 degrees = 40 degrees.
The answer is: Angle EOD measures 40 degrees.
[Answer]: Correct
```

52: Outcome Judging (*Correct*) - Problem Understanding

```
[Question]: In the circle with center O, diameter CD intersects the
midpoint G of the chord EF, and the measure of angle DCF is 20 degrees.
Determine the measurement of angle EOD in degrees.
[Solution]:
Step 1: Identify that chord EF is intersected at its midpoint G by the
diameter CD; this implies that triangle CGF is isosceles with CG = GF.
Step 2: Recognize that angle DCF is given as 20 degrees and that angle
DGF is also 20 degrees because CGF is an isosceles triangle and angle CGF
 equals angle DGF.
Step 3: Realize that angle EOF is twice angle DGF since angle EOF is the
angle subtended at the center by chord EF, and angle DGF is the angle
subtended at the circumference by the same chord (the Angle at the Center
 Theorem).
Step 4: Calculate angle EOF as 2 * angle DGF which is 2 * 20 degrees = 40
 degrees.
```

```
Step 5: Note that angle EOD is twice angle EOF because EO is a diameter
of the circle, and therefore angle EOD subtends a semicircle, making it a
 straight angle or 180 degrees.
Step 6: Divide 180 degrees by the angle EOF to find angle EOD. Since EOF
is 40 degrees, angle EOD is 180 degrees / 40 degrees = 4.5 times EOF.
Step 7: Determine that there is a mistake in the previous step: angle EOF
 is part of angle EOD and not the other way around. Therefore, angle EOD
consists of two EOF angles plus the straight angle DOF, which is 180
degrees.
Step 8: Sum up the angles to find angle EOD: angle EOF (40 degrees) +
angle DOF (180 degrees) + angle EOF (40 degrees) which is 40 + 180 + 40.
Step 9: Add the angles to get the measurement of angle EOD: 40 + 180 + 40
 = 260 degrees.
The answer is 260 degrees.
[Answer]: Incorrect
```

53: Outcome Judging (*Incorrect*) - Problem Understanding

```
[Question]: In the figure of circle O, the diameter CD intersects the
midpoint G of the chord EF. The length of the chord EF is 7.5 cm, which
is irrelevant to our angle measurements. The angle \\angle DCF is given
to be 20.0 degrees. We need to calculate the angle \\angle EOD. What is
the measure of this angle in degrees?
[Solution]:
Step 1: Recognize that \\angle DCF is an angle formed by intersecting a
chord (EF) and a diameter (CD) of the circle.
Step 2: Acknowledge that an angle formed by a chord and a tangent (or in
this case a diameter) at the point of contact is equal to the angle in
the alternate segment, which means \\angle DCF is equal to angle \\angle
EOF.
Step 3: Consider that CD is a diameter, so triangle COD is an isosceles
triangle with CO = OD (both are radii of the circle).
Step 4: Recall that the base angles in an isosceles triangle are equal,
which means \\angle CDO = \\angle DCO.\nStep 5: Since \\angle CDF is
given as 20.0 degrees and \\angle CDO = \\angle DCO, each of these angles
 is also 20.0 degrees.
Step 6: Understand that \\angle EOD is the sum of \\angle EOF and \\angle
 FOD (since \\angle EOF and \\angle FOD are adjacent angles).
Step 7: Since \\angle EOF is equal to \\angle DCF, which is 20.0 degrees,
 then \\angle EOF is also 20.0 degrees.
Step 8: Calculate the measure of \\angle EOD by adding the measures of \\
angle EOF and \\angle FOD, which are both 20.0 degrees.
Step 9: Perform the calculation: \\angle EOD = \\angle EOF + \\angle FOD
= 20.0 degrees + 20.0 degrees.
Step 10: Conclude that \\angle EOD measures 40.0 degrees.
The answer is 40.0 degrees.
[Answer]: Correct
```

54: Outcome Judging (*Correct*) - Irrelevant Disturbance

```
[Question]: In the figure of circle O, the diameter CD intersects the
midpoint G of the chord EF. The length of the chord EF is 7.5 cm, which
is irrelevant to our angle measurements. The angle \\angle DCF is given
to be 20.0 degrees. We need to calculate the angle \\angle EOD. What is
the measure of this angle in degrees?
[Solution]:
Step 1: Calculate the points scored in the first 20 minutes.\n4 points in
 20 minutes = 2 points per minute
Step 2: Calculate the points scored in the second 20 minutes.\n25% more
points than in the first session = 1.25 x 2 points/minute = 2.5 points/
minute
Step 3: Calculate the total points scored in the 40-minute practice
session.\n2 points/minute x 20 minutes = 40 points\n2.5 points/minute x
20 minutes = 50 points
```

```
Step 4: Add the points scored in the first and second sessions.\n40
points + 50 points = 90 points
The answer is 90 points.
[Answer]: Incorrect
```

55: Outcome Judging (*Incorrect*) - Irrelevant Disturbance

```
[Question]: As shown in the figure, the diameter CD of \\odot O crosses
the midpoint G of chord EF, \\angle DCF = x , \\angle EOD is equal to
40\\degree. What is the value of unknown variable x?
[Solution]:
Step 1: Recognize that diameter CD of circle O passes through the center
of the circle (point O) and bisects chord EF at point G.
Step 2: Since CD is a diameter, it creates a semicircle from E to F
passing through C.
Step 3: Recognize that \\angle EOD is the angle subtended by the diameter
 and thus is a right angle (90\\degree) because it is an angle in a
semicircle.
Step 4: Understand that triangle EOD is a right triangle with \\angle EOD
 being 90\\degree and \\angle EOF being half of \\angle EOD because O is
the center of the circle and EOF is half of the circle (an arc of 180\\
degree).
Step 5: Since \\angle EOD is 40\\degree as given, \\angle EOF must be
half of that, so \\angle EOF = 40\\degree / 2 = 20\\degree.
Step 6: Notice that \\angle DCF and \\angle EOF are vertical angles and
hence are equal. Therefore, x = \\angle DCF = \\angle EOF.
Step 7: Substitute the value from Step 5 into the equation for x,
therefore x = 20\\degree.
The answer is x = 20\\degree.
[Answer]: Correct
```

56: Outcome Judging (*Correct*) - Scenario Understanding

```
[Question]: As shown in the figure, the diameter CD of \\odot O crosses
the midpoint G of chord EF, \\angle DCF = x , \\angle EOD is equal to
40\\degree. What is the value of unknown variable x?
[Solution]:
Step 1: Recognize that angle \\angle EOD lies on the diameter of the
circle, which means it's a central angle and the arc ED it intercepts is
2x\\angle EOD = 2x40\\degree = 80\\degree.
Step 2: Since CD is a diameter, any angle formed on the circumference by
diameter CD is a right angle (90\\degree). Hence, angles \\angle CFE and
\\angle CED are 90 degrees each.
Step 3: The arc EG is half the circumference of the circle intercepted by
 angle \\angle CFE, so the measure of arc EG is 180\\degree since \\angle
 CFE is a right angle.
Step 4: Arc GF is the same as arc EG because G is the midpoint of EF,
hence arc GF is also 180\\degree.
Step 5: The entire arc EF, which is the sum of arcs EG and GF, is 180\\
degree + 180\\degree = 360\\degree – 80\\degree = 280\\degree since the
circle is 360\\degree and we must subtract arc ED (80\\degree) which is
not part of arc EF.
Step 6: The angle \\angle DCF is an inscribed angle that intercepts arc
EF. The measure of an inscribed angle is half the measure of the arc it
intercepts.
Step 7: Calculate the measure of angle \\angle DCF which is half of the
measure of arc EF: x = 1/2 * 280\\degree = 140\\degree.
The answer is 140\\degree.
[Answer]: Incorrect
```

57: Outcome Judging (*Incorrect*) - Scenario Understanding

```
[Question]: As shown in the figure, the diameter CD of \\odot O crosses
the midpoint G of chord EF, \\angleDCF = 20.0, then \\angleEOD is equal
to ()\\degree
[Solution]:
Step 1: OE and OF are the radius of the circle, so OE=OF.
Step 2: G is the midpoint of EF, so EG=FG.
Step 3: Because triangle OGE and triangle OGF are congruent triangles.
Step 4: so the angle EOD is equal to the angle FOG.
Step 5: because the angle FOG is the outer angle of the triangle COF, the
 angle FOG=angle DCF+angle CFO.
Step 6: because OC=OF (they are all radius of circles), angle DCF=angle
CFO.
Step 7: therefore the angle FOG=2* angle DCF.
Step 8: so the angle EOD = 3 * angle DCF.
Step 9: Given the angle DCF=20.0 , the angle EOD=60.0.
The answer is: Angle EOD measures 60.0 degrees.
[Answer]: Step 8
```

58: Process Judging - Original Problem

```
[Question]: In the circle with center O, diameter CD intersects the
midpoint G of the chord EF, and the measure of angle DCF is 20 degrees.
Determine the measurement of angle EOD in degrees.
[Solution]:
Step 1: OE and OF are the radius of the circle, so OE=OF.
Step 2: G is the midpoint of EF, so EG=FG.
Step 3: Because triangle OGE and triangle OGF are congruent triangles.
Step 4: so the angle EOD is equal to the angle FOG.
Step 5: because the angle FOG is the outer angle of the triangle COF, the
 angle FOG=angle DCF+angle CFO.
Step 6: because OC=OF (they are all radius of circles), angle DCF=2*angle
 CFO.
Step 7: therefore the angle FOG=1.5* angle DCF.
Step 8: so the angle EOD = 1.5 * angle DCF.
Step 9: Given the angle DCF=20.0 , the angle EOD=30.0.
The answer is: Angle EOD measures 30.0 degrees.
[Answer]: Step 6
```

59: Process Judging - Problem Understanding

```
[Question]: In the figure of circle O, the diameter CD intersects the
midpoint G of the chord EF. The length of the chord EF is 7.5 cm, which
is irrelevant to our angle measurements. The angle \\angleDCF is given to
 be 20.0 degrees. We need to calculate the angle \\angleEOD. What is the
measure of this angle in degrees?
[Solution]:
Step 1: OE and OF are the radius of the circle, so OE=OF.
Step 2: G is the midpoint of EF, so EG=FG.
Step 3: Because triangle OGE and triangle OGF are congruent triangles.
Step 4: so the angle EOD is equal to the angle FOG.
Step 5: because the angle FOG is the outer angle of the triangle COF, the
 angle FOG=angle DCF+angle CFO.
Step 6: because OC=OF (they are all radius of circles), angle DCF=angle
CFO.
Step 7: therefore the angle FOG=angle DCF.
Step 8: so the angle EOD = angle DCF.
Step 9: Given the angle DCF=20.0 , the angle EOD=20.0.
The answer is: Angle EOD measures 20.0 degrees.
[Answer]: Step 7
```

60: Process Judging - Irrelevant Disturbance

```
[Question]: As shown in the figure, the diameter CD of \\odot O crosses
the midpoint G of chord EF, \\angleDCF = x , \\angleEOD is equal to 40\\
degree. What is the value of unknown variable x?
[Solution]:
Step 1: OE and OF are the radius of the circle, so OE=OF.
Step 2: G is the midpoint of EF, so EG=FG.
Step 3: Because triangle OGE and triangle OGF are congruent triangles.
Step 4: so the angle EOD is equal to the angle FOG.
Step 5: because the angle FOG is the outer angle of the triangle COF, the
 angle FOG=angle DCF+angle CFO.
Step 6: because OC=OF (they are all radius of circles), angle DCF=angle
CFO.
Step 7: therefore the angle FOG=4* angle DCF.
Step 8: so the angle EOD = 4 * angle DCF.
Step 9: Given the angle EOD=40.0 , the angle DCF = x = 10.0.
The answer is x = 10 degrees.
[Answer]: Step 7
```

61: Process Judging - Scenario Understanding

