# OpenReview forum: "Is Your Model Really A Good Math Reasoner? Evaluating Mathematical Reasoning with Checklist"
_ICLR.cc/2025/Conference — ICLR 2025 Poster_

### Official Review · Reviewer_n8tf · 2024-10-26

**Soundness:** 3
**Presentation:** 4
**Contribution:** 2
**Rating:** 5
**Confidence:** 4

**Summary:**

This paper explores the question that how to comprehensively evaluate LLMs’ mathematical abilities. Considering that if a model understands a problem, it should be applied across different tasks and problem varieties, they introduce MATHCHECK. MATHCHECK is a checklist for testing task generalization and reasoning robustness. For textual and multimodal language models, they develop MATHCHECK-GSM and MATHCHECK-GEO separately utilizing MATHCHECK.

**Strengths:**

1.	The paper studies an important research question from a fresh perspective, evaluating the models’ reasoning abilities across various tasks and problem varieties.
2.	Abundant experiments: A total of 43 models are evaluated on the proposed benchmarks, which serve as comprehensive baselines for further studies.
3.	The paper is well-written and easy to follow.

**Weaknesses:**

1.	Although the paper attempts to analyze the mathematical reasoning capabilities of existing models from multiple aspects (including PS: Problem Solving, AJ: Answerable Judging, OJ: Outcome Judging, PJ: Process Judging, OP: Original Problem, PU: Problem Understanding, ID: Irrelevant Disturbance, SU: Scenario Understanding), there are the following shortcomings. On one hand, some models (such as O1-preview) have already achieved very good performance (>90%) on these datasets and have shown extremely strong robustness, indicating that the evaluation methods proposed in the paper may not be challenging enough for the most advanced models currently available. Moreover, the reviewer did not find any new insights proposed by the evaluation method for the improvement of existing models. On the other hand, the goal of the paper is to explore whether models truly possess mathematical reasoning abilities, and the multiple aspects proposed seem insufficient: if one model performs well in all the proposed aspects, does it truly have reasoning abilities?
2.	Given that these benchmarks are generated by LLMs and only verified by humans, how to ensure the diversity across different question groups? For example, with different seed questions, LLMs may tend to include similar irrelevant information when generating the “irrelevant disturbance” variety, leading to a less comprehensive evaluation.

**Questions:**

1.	A seed problem is expanded to include 4 math tasks and 4 problem varieties (as shown in Fig.1), resulting in a total of 4\*4=16 problems. Therefore, for the 129 style groups in MATHCHECK-GSM, the total number of samples should be 129\*16=2064. However, in Sec. 3.1, the number of samples in MATHCHECK-GSM is reported as 3096. Could you clarify this discrepancy?

---

> ### Author Response · Authors · 2024-11-21
> **Response to Reviewer n8tf**
>
> Thank you for reviewing our work. Below, we will address your concerns point by point:
>
> > The evaluation methods proposed in the paper may not be challenging enough for the most advanced models currently available.
>
> **The goal of MathCheck is to provide a more comprehensive evaluation of mathematical reasoning abilities at a given difficulty level, rather than simply making math problems harder to stump the model**. To this end, we design a test checklist from both reasoning robustness and task generalization. It allows users to better evaluate the math reasoning ability and conducts fine-grained analysis. Besides, it is a plug-and-play testing framework suitable for mathematical problems of different difficulty.
>
> We construct MathCheck-GSM for two motivations: (1) validating the effectiveness of MathCheck paradigm. (2) determining whether LLMs are capable of reasoning at the grade school level while most of them achieve above 85% on GSM8k before. **Through our experiments, we observe significant performance drops in many LLMs, which validates the effectiveness of our paradigm and reveals that some LLMs rely on memorization to solve GSM8k problems.**
>
> Meanwhile, we also want to evaluate models on MathCheck with multimodal math problems. To address this, we developed **MathCheck-Geo comprising high-school geometry problems, on which GPT-4o achieves only 65.3% and we successfully identified weaknesses (robustness or task generalization) in some MLLMs.**
>
> > Not find any new insights proposed by the evaluation method for the improvement of existing models.
>
> Evaluation is of significant importance to assist the research and development of LLMs. Our benchmark offers several insights w.r.t. analyzing and improving the mathematical reasoning abilities of LLMs in the following aspects:
>
> 1. **Fine-grained analysis**: MathCheck offers various robustness variants and task types, enabling users conduct fine-grained analysis on their models and target weaknesses such as robustness. It helps further improve the general math reasoning abilities of models
> 2. **Identification of pre-training generalization**: Most base models exhibit strong reasoning consistency on MathCheck, indicating that the mathematical abilities learned from pretraining are generalized and robust.
> 3. **Insights on fine-tuning and data augmentation**: Figure 5 implies that finetuning solely on massive problem-solving data is not the right direction to improve general math reasoning abilities, highlighting the need for more diverse and high-quality SFT data.
>
> >  The multiple aspects proposed seem insufficient: if one model performs well in all the proposed aspects, does it truly have reasoning abilities?
>
> In order to make the problem variants comprehensive, we propose to **construct test samples from reasoning robustness and task generalization**. In reasoning robustness, we consider Problem Understanding, Irrelevant Disturbance, and Scenario Understanding. In task generalization, we consider answerability detection(Answerable Judging) , answer correctness judgment (Outcome Judging) , process-error identifying (Process Judging). **These problem variants comprehensively test multiple reasoning skills in mathematical reasoning**. Compared to previous evaluation paradigms, we make a big step forward: MathCheck is **more closer** mathematical intelligence, which we have validated in Section 3.4 through correlation with Private Data and Compression Efficiency.
>
> > How to ensure the diversity across different question groups? For example, with different seed questions, LLMs may tend to include similar irrelevant information when generating the “irrelevant disturbance” variety, leading to a less comprehensive evaluation.
>
> **We have considered diversity during the generation process.**
>
> - When generating Irrelevant Disturbance question variants, LLM creates distractors related to the original question's topic, ensuring the irrelevant information is unique. For instance, in the original question:
>  _”A robe takes 2 bolts of blue fiber and half that much white fiber. How many bolts in total does it take?”_
> its variant is:
> _“A tailor is crafting a luxurious robe. The design requires 2 bolts of blue fiber and half that amount of white fiber. **To add grandeur, the tailor also considered using 3 bolts of golden thread from the sun's rays, but eventually decided it would be too gaudy for the ceremony.** How many bolts in total are needed for the robe, disregarding the golden thread?”_
> We can see that the distractor is topic-related.
>
> - Furthermore, when generating binary-classification tasks (Answerable Judging and Outcome Judging), we created both a positive and a negative sample for each problem.
> - For Process Judging task, we instruct the LLMs to generate error of given a random step rather than fixed step, ensuring diversity in the generated error.

---

> ### Author Response · Authors · 2024-11-21
> **Further response to Reviewer n8tf**
>
> > In Sec. 3.1, the number of samples in MATHCHECK-GSM is reported as 3096. Could you clarify this discrepancy?
>
> In each group, since answerable judging and outcome judging are binary-classification tasks, we include two different labels in these units for fair evaluation. For example, in each answerable-judging unit, we have 2 samples (Answerable and Unanswerable). This distribution ensures that our data is balanced and diverse. **Therefore, a seed problem is expanded to 4+4$\times\$2+4$\times\$2+4 =24 problems. For the 129 style groups in MATHCHECK-GSM, the total number of samples is 129$\times\$24=3096.** In the original version, we have provided relevant descriptions in Appendix C.1 and Table 5, please check.
>
> Thanks for your review again. We sincerely hope that our responses can address your concerns and contribute to a better evaluation on our work.

---

> ### Author Response · Authors · 2024-11-26
> **Response to Reviewer n8tf**
>
> Dear Reviewer,
>
> We sincerely appreciate your feedback and effort. During the rebuttal process, we have earnestly addressed your concerns and questions with detailed, point-by-point replies, including:
>
> 1. The number of samples in MathCheck-GSM.
> 2. The effectiveness of MathCheck.
> 3. Insights for improving general mathematical reasoning.
> 4. The diversity of generated data.
>
> We sincerely hope that our responses could satisfactorily address your questions and concerns. If you have any further inquiries or require additional clarification, please do not hesitate to let us know; we remain available and willing to address any additional questions or provide further modifications as needed.
>
>
> Once again, thank you for your valuable input!
>
>
> Best wishes,
> Authors of Paper 7051

---

### Official Review · Reviewer_GGUq · 2024-10-29

**Soundness:** 3
**Presentation:** 3
**Contribution:** 3
**Rating:** 6
**Confidence:** 3

**Summary:**

This paper introduces MATHCHECK, a new evaluation framework for assessing mathematical reasoning capabilities in LLMs. The framework's key innovation is its comprehensive evaluation approach that tests both task generalization and reasoning robustness. The authors develop two datasets using this framework: MATHCHECK-GSM for text based mathematical reasoning and MATHCHECK-GEO for multimodal geometric reasoning. They demonstrate that while some frontier models like GPT-4o maintain strong performance across their checklist, many other models show significant performance degradation compared to their standard problem solving abilties. The authors also provide evidence that MATHCHECK correlates better with true mathematical reasoning abilities compared to other benchmarks. Finally, the authors show that the current paradiem of finetuning models for math probalem solving capabilities leads to stagnation or even degredation in other areas of math reasoning.

**Strengths:**

In terms of originality and quality, it introduces a novel evaluation paradigm that goes beyond traditional problem-solving assessments to comprehensively evaluate mathematical reasoning through multiple tasks and robustness tests. While there have been some previous work on creating function variations, this paper goes beyond simply rewritting the question.
The methodology is rigorous, with extensive empirical validation involving 43 models and a careful data generation process that achieves an 84.61% pass rate with manual validation. The authors validate their framework's effectiveness through correlation with private data and compression efficiency metrics, imlying performance on this benchmark is more robustly capturing llm math abilities.
The paper is reasonably clear in its presentation, with well-structured explanations of the framework and comprehensive documentation of methodology.
The paper is moderately siginificant. The paper provides a more reliable way to evaluate true mathematical reasoning capabilities of LLMs and is exstensible to other domains as well. In particular, the ability to evauate llms on areas other than problem solving is very useful to MCTS and LLM-as-a-judge methods.

**Weaknesses:**

Would recommend citing 1+ papers on functional variations benchmarks which programatically generate question variations.

MATHCHECK-GSM seems to be saturated by frontier models, making it hard to get an accurate measure of their relative abilities. Constructing a MATHCHECK benchmark from MATH or another harder benchmark would be ideal.

In Appendix C, it is unclear what is meant by "data statistic". I assume in tables 3 and 4 it is the total count of questions for that variation type, but I'm not sure what table 5 is trying to show.

Recommended:
- Evaluate models on MATHCHECK adapted to other reasoning tasks (i.e. commonsense reasonin and code)

Minor:
- Use (M)LLM and MLLM inconsistently/unclear what the term means
- It would be nice to have tables 1 and 2 in a plot somehow so we can better see how performance compared across evaluation categories.
- The text in Figure 5 is a bit small and in general hard to interpret
   - I would recommend using a bar chart or similar where it is easier to see the performance deltas (this might require aggregating over the rows)
- Figure 6 should have an x-axis label
- In figure 7, the pink and red are hard to differenciate (perhaps remove the color fills and only include the outlines)

**Questions:**

- What is the pass rate for GPT-4o rewritting for the MATHCHECK-GEO benchmark?
- The data generation pipeline has a pass rate of ~85%. It is not explicit in the paper whether the released banchmark has filtered out the questions that don't pass or left them in.

---

> ### Author Response · Authors · 2024-11-21
> **Response to Reviewer GGUq**
>
> Thanks for your time and valuable comments. Below, we will address your concerns and suggestions point by point:
>
> > Citing functional variations benchmarks.
>
> Indeed, the Functional Variations Benchmarks are highly relevant to our work. We have now cited two studies on Functional Variations Benchmarks in the Related Work section [1][2].
>
> Reference:
>
> [1] Functional benchmarks for robust evaluation of reasoning performance, and the reasoning gap. (Srivastava et al., Arxiv 2024)
>
> [2] Putnam-axiom: A functional and static benchmark for measuring higher level
> mathematical reasoning. (Aryan Gulati et al., NeurIPS 2024 MATH Workshop)
>
> > Performance of MathCheck-GSM.
>
> **The goal of MathCheck is to provide a more comprehensive of mathematical reasoning abilities at a given difficulty level, rather than simply making math problems harder to stump the model.** Base on this, we design a test checklist from reasoning robustness and task generalization. It allows users to better evaluate the math reasoning ability and conducts fine-grained analysis. Besides, it is a plug-and-play testing framework suitable for mathematical problems of different difficulty.
>
> We construct MathCheck-GSM for two motivations: **(1) validate the effectiveness of MathCheck paradigm. (2) determine whether LLMs are capable of reasoning at the grade school level while most of them achieve above 85% on GSM8k before.** Through our experiments, we observe significant performance drops in many LLMs, which validates the effectiveness of our paradigm and reveals that some LLMs rely on memorization to solve GSM8k problems.
>
> Meanwhile, we also want to evaluate models on MathCheck with multimodal math problems. To address this, we developed MathCheck-Geo comprising **high-school geometry problems , on which GPT-4o achieves only 65.3% and we successfully identified weaknesses (robustness or task generalization.) in some MLLMs.**
>
> **Through above datasets, the effectiveness and adaptability of MathCheck are validated.** Recently, we also recognize the community's eagerness in more challenging textual mathematical problems in MathCheck paradigm. In the future work, we will develop MathCheck with more difficult textual benchmark such as MATH. Thanks for your suggestion!
>
> > Data statistic.
>
> Sorry for the confusion. **Table 5 shows the variants count of each seed question.** In instance, for a seed question from gsm8k, we have 1 PU-PS vaiants and 2 PU-AJ vaiants (Answerable and Unanswerable respectively). This distribution ensures that our data is balanced and diverse. We will make the descriptions clearer to avoid confusion.
>
> > Minor modifications.
>
> Thanks for your suggestions. In our paper, MLLM means multimodal LLMs while (M)LLM means multimodal LLMs and LLMs. **Based on your suggestions, we have made several modifications, including:**
>
> 1. Combine Table 1 and Table 2 in same page for better comparison.
> 2. Use a bar chart in Figure 5 in order to easily see the performance delta.
> 3. Add x-axis label in Figure 6.
> 4. Remove the color fills in Figure 7.
>
> > Pass rate of MathCheck-Geo.
>
> Regrettably, we did not record the pass rates for MathCheck-Geo. However, **the same as MathCheck-GSM, we manually corrected all failed generated data in MathCheck-Geo ensuring that no error data are included in the evaluation.** Therefore, our MathCheck-GSM and MathCheck-Geo are entirely correct and the evaluation results are reliable.
>
> > Recommended: Evaluate models on MATHCHECK adapted to other reasoning tasks (i.e. commonsense reasoning and code)
>
> In Section 5, we discussed the potential application of MathCheck to other reasoning tasks and provided examples in commonsense reasoning and code generation. However, we did not scale these cases and evaluate them due to the cost. Actually, we conducted a preliminary test using GPT-4o-mini to answer commonsense reasoning variants and observed that it was indeed affected:
>
> **Original Question**：Yesterday‘s date was 4/30/2021. What is the date tomorrow in MM/DD/YYYY?
>
> Ground Truth: 5/2/2021
>
> Predict: Tomorrow's date will be 05/02/2021. **(True)**
>
> **Irrelevant Disturbance-Problem Solving variants**: Yesterday was April 30, 2021. A week ago it was 4/23/2021. What is the date tomorrow in MM/DD/YYYY?
>
> Ground Truth: 5/2/2021
>
> Predict: Tomorrow's date will be 05/01/2021 **(False)**
>
> **Irrelevant Disturbance-Outcome Judging variants**: Yesterday was April 30, 2021. A week ago it was 4/23/2021. What is the date tomorrow in MM/DD/YYYY?  "solution": The date tomorrow will be 05/01/2021.
>
> Ground Truth: Incorrect
>
> Predict: Yes, the solution is correct. If yesterday was April 30, 2021, then tomorrow will be May 1.
> 2021. In MM/DD/YYYY format, that is 05/01/2021.  **(False)**
>
> We observe that GPT-4o-mini successfully answers the original question but failed to respond correctly to two variants. **This highlights that reasoning consistency issues are prevalent across various reasoning tasks. MathCheck paradigm can effectively reveal such problems.**

---

> ### Author Response · Authors · 2024-11-21
> **Further response to Reviewer GGUq**
>
> > Whether the released banchmark has filtered out the questions?
>
> Yes. We manually checked all of generated data in MathCheck-GSM and MathCheck-Geo. **After that, annotators corrected each failed data. Therefore no wrong generated data are included in evaluation.** We include the detailed description of human evaluation process in the Appendix C.3, please check.
>
>
> Thanks for your review again. If our response can alleviate your concerns and promote your positive view of the paper, we would appreciate it if you could strengthen the recommendation.

---

> > ### Author Response · Authors · 2024-11-26
> > **Response to Reviewer GGUq**
> >
> > Dear Reviewer,
> >
> > We sincerely appreciate your feedback and the time you have dedicated to reviewing our paper. In our rebuttal, we have earnestly addressed your concerns and questions, and provided detailed responses, including:
> >
> > - Introducing functional variations benchmarks in the Related Work.
> > - Refining the paper, including merging Table 1 and Table 2, modifying Figure 5 to a bar chart, and making adjustments to Figures 6 and 7.
> > - Clarifying the performance of MathCheck.
> > - Case study of MathCheck on commonsense reasoning tasks.
> > - Further clarification on our question filtering process.
> >
> > More detailed updates can be found in blue highlights in our revised paper. We appreciate your insightful suggestions, which have enhanced the clarity and comprehensibility of our work.
> >
> > We hope that our revisions and explanations adequately address your concerns. If you have any additional questions or require further clarification, please do not hesitate to reach out. We remain fully available and eager to provide further modifications as needed.
> >
> > Best wishes,
> > Authors of Paper 7051

---

> > > ### Comment · Reviewer_GGUq · 2024-11-27
> > >
> > > Thank you for the detailed responses. I still believe a version of the MATH dataset would improve this paper, but I will keep my score as is.

---

> ### Author Response · Authors · 2024-12-03
> **Response to Reviewer-GGUq:**
>
> Dear Reviewer-GGUq:
>
> Thanks for your review and positive score.
>
> As our main contribution, MathCheck is a general paradigm for comprehensive mathematical reasoning evaluation and fine-grained analysis of models. We validated its effectiveness and adaptability on MathCheck-GSM(text, simple question) and MathCheck-Geo(multi-modal, high school question).In our experiments, we choose GSM8k instead of MATH because many math reasoning studies conducted on GSM8K and one specific study [1] using the private dataset GSM1K, which can further verify our analysis.
>
> As a textual math problem dataset, the process of transform MATH into MathCheck paradigm is similar to GSM8k since it primarily utilizes GPT-4o's semantic understanding and rewriting capabilities rather than problem-solving abilities. For your reference, we collected MathCheck data in MATH problems to demonstrate its transferability. Specifically, we utilized GPT-4o as rewritten model for data generation and manually checked each of them through human verification process. We find that the pass rate checked by human annotators is similar with those on GSM. Due to limited time, we collect 240 samples, and you can check these reference samples on https://anonymous.4open.science/r/MathCheck/MATH_checklist.json.
>
> Thank you once again for your review and valuable suggestions. Hopefully, this can solve your concern.
>
> [1] [A careful examination of large language model performance on grade school arithmetic](https://arxiv.org/abs/2405.00332) (Zhang et.al, NeurIPS 2024)
>
> best wishes,
> Authors of Paper 7051

---

### Official Review · Reviewer_WuK6 · 2024-11-03

**Soundness:** 3
**Presentation:** 3
**Contribution:** 3
**Rating:** 6
**Confidence:** 4

**Summary:**

This paper introduces MathCheck, a new mathematical reasoning benchmark for large language models (LLMs). It aims to address the limitations of existing benchmarks, which often focus on evaluating individual tasks rather than LLM models' understanding of the problems. MathCheck incorporates carefully augmented math problems in different dimensions, including generalization and reasoning robustness.

**Strengths:**

The paper provides a novel approach to evaluating math problems by considering whether the models understand the problems, not just solve them. The dimension of task generalization is innovative, offering various methods to judge problem answers and providing insights into evaluating reasoning ability. The linearity and performance on different complexity levels demonstrate the potential of the new benchmark for comprehensive evaluation of models' math reasoning abilities.

**Weaknesses:**

The data generation pipeline with an 86% pass rate, although good, doesn’t sound ideal given the scale of samples. The paper could benefit from briefly discussing in the main text when showing the pass rate how low-quality generated questions are handled when detected.

Nitpicks: Line 431: It would be helpful to include references to the alleged works. Line 463: quote left mark is incorrect.

**Questions:**

In Figure 6, could the accuracy drop be attributed to the increasing difficulty LLMs faced in generating or reconstructing task generation entries for more complex problems?

Appendix C2 presents pass rates for different rewriting types. Would it be practical to also show the pass rates for different complexity levels? (This may not be necessary if the such questions are omitted in evaluation.)

---

> ### Author Response · Authors · 2024-11-21
> **Response to Reviewer WuK6**
>
> Thank you for reviewing our work. Below, we will address your concerns and suggestions point by point:
>
> > How to handle the low-quality generated data?
>
> We manually checked all of generated data in MathCheck-GSM and MathCheck-Geo. After that, **annotators corrected each failed data**. **This approach ensures our MathCheck-GSM and MathCheck-Geo are entirely correct and the evaluation results are reliable.** We include the detailed description of human evaluation process in the Appendix C.3, please check.
>
> > Line 431: It would be helpful to include references to the alleged works. Line 463: quote left mark is incorrect.
>
> Thank you. **We include related references to the alleged works and correct the typo(quote left mark) in the new version**, please check.
>
> > In Figure 6, could the accuracy drop be attributed to the increasing difficulty LLMs faced in generating or reconstructing task generation entries for more complex problems?
>
> All of our test data has been manually reviewed and corrected to ensure the reliability of the evaluation. Therefore,  the performance drop in Figure 6 is not due to generated data correctness of difficult problems. **Under such fair comparison, the performance drop implies MathCheck better demonstrates the reasoning skills and capabilities required when problems become difficult.**
>
> > Pass rates of different complexity questions.
>
> We did not record the pass rates for different complexity levels. However, **we manually corrected all failed generated data ensuring that no error data are included in the evaluation.**
>
> Thanks for your review again! We hope our responses can address your concerns.

---

> ### Author Response · Authors · 2024-11-26
> **Response to Reviewer WuK6**
>
> Dear Reviewer,
>
> We sincerely appreciate your feedback and the time you have dedicated to reviewing our paper. During the rebuttal process, we have earnestly addressed your concerns and questions with detailed, point-by-point replies, including:
>
> - The strategy of handling incorrect generated data.
> - Introducing references in Section Behavior of Math Models.
> - Typo correction.
> - The reason of performance drop in Figure 6.
> - Clarifying the filtering process of generated data from complexity questions.
>
> Detailed updates are highlighted in blue in our revised paper. We appreciate these valuable suggestion, which enhanced the clarity and comprehensibility of our work.
>
> We hope that our response and revision adequately address your concerns. If you have any further questions or require additional clarification, please feel free to reach out. We remain fully available and happy to provide further modifications as needed.
>
> Best wishes,
> Authors of Paper 7051

---

> > ### Comment · Reviewer_WuK6 · 2024-11-28
> >
> > Thank you for addressing the questions thoroughly. I appreciate the authors' efforts in ensuring the accuracy of all generated data. I am satisfied with the response provided and will maintain a positive rating.

---

> > > ### Author Response · Authors · 2024-12-03
> > > **Response to Reviewer-WuK6:**
> > >
> > > Dear Reviewer-WuK6:
> > >
> > > Thanks for your review and satisfaction with our response. We also appreciate your suggestions, which helped improve the quality of our paper. Thank you once again for your valuable feedback and positive score.
> > >
> > > best wishes,
> > > Authors of Paper 7051

---

### Official Review · Reviewer_eZPq · 2024-11-04

**Soundness:** 3
**Presentation:** 4
**Contribution:** 3
**Rating:** 8
**Confidence:** 4

**Summary:**

This paper presents MathCheck, a benchmark/checklist for testing (M)LLMs' task generalization and reasoning robustness in mathematical problems. It utilizes GSM8k and GeoQA to create new math problems that are of different styles (e.g., irrelevant disturbance) and different types (e.g., process judging). By evaluating 26 LLMs and 17 MLLMs on this benchmark, the authors find that some models show generally robust performance (e.g., strong close models like GPT-4o), while other models (e.g., Qwen1.5-72B-Chat) show performance drop in alternative problem settings (e.g., OJ and ID). Overall for MLLMs there is large room for improvement on the multi-modal subset (GEO) of the benchmark. They include several analyses to support the reliability of the benchmark and show how the main ideas in the paper can be applied to other evaluation domains such as commonsense reasoning.

**Strengths:**

- The paper is clearly written. The examples and figures are illustrative. For example, I can easily understand what the benchmark is about by looking at Figure 1. The sections are structured well.

- The task setup is innovative (i.e., the 4 x 4 matrix). It is valuable to have these different styles and formats of problems in one place for evaluation and comparison. The argument that the 4 x 4 setup is relevant to measuring generalization and robustness (at least to some degree) is sound.

- The benchmark has text only and multi-modal parts, which are of interest to multiple sub-communities in the field.

- The authors studied a comprehensive set of LLMs, covering models that are open and closed, big and small, generalist and specialized. And they studied several different prompting strategies, which we know matter for mathematical reasoning.

- The authors do a good job attempting to ensure data quality and establish benchmark reliability. For example, they show human pass rates w.r.t. LLM-based problem generation, and they show that the accuracy of MathCheck-GSM is highly correlated to that of GSM1k, a putative high-quality private dataset.

- The authors not only discuss how the evaluation method introduced in this paper can be general (applied to other domains), but they also have already done work in that direction by creating variants of "data understanding" and code generation tasks. This would prompt other researchers to consider how this method may be relevant to their own domains of interest.

Overall, I believe that this paper and dataset are a valuable contribution to the AI for math research community.

**Weaknesses:**

- One weakness of the benchmark is about the difficulty of the problems. MathCheck-GSM is based on GSM8k, which is a grade-school-level benchmark that the SoTA models perform very well on. And correspondingly they tend to perform well on MathCheck-GSM. It would have been a stronger benchmark if MathCheck also included high-school and beyond problems, so that we could have a better understanding of where and how models like GPT-4o and O1-preview fail. But this is not a major weakness and can be left for future work. MathCheck already has promise for measuring progress in smaller/weaker models and multi-modal models.

- A similar weakness to the above is about the diversity of problems. The seed data only contain two types of problems: simple word problems and geometry problems.

- The related work section needs to be improved/expanded. Only talking about other math benchmarks is not enough. I encourage the authors to add at least two paragraphs. One can be on what people have done to improve mathematical reasoning with LLMs (training, inference, prompting, LLMs + formal tools, etc.)—the broad landscape of AI for math. Another can be similarly spirited benchmarks in other domains. For example, the use of counterfactual tasks in [1] is also for evaluating how robust and general LLMs' problem understanding and reason abilities are.

[1] Wu et al. (2024) "Reasoning or reciting? Exploring the capabilities and limitations of language models through counterfactual tasks" NAACL.

**Questions:**

Who are the human annotators that evaluated the quality of GPT-4T rewriting? Are they (a subset of) the authors? (That is implied from C.3, but I think it should be explicitly stated.) How did one determine whether a problem "passed"? I just encourage the authors to add a few sentences to better clarify the human evaluation process.

---

> ### Author Response · Authors · 2024-11-21
> **Response to Reviewer eZPq**
>
> Thanks for your time and valuable comments. Below, we address your concerns and suggestions point by point:
>
> > About the seed question GSM8k and Geometry-QA.
>
> **The goal of MathCheck is to provide a more comprehensive of mathematical reasoning abilities at a given difficulty level, rather than simply making math problems harder to stump the model.** Base on this, we design a test checklist from reasoning robustness and task generalization. It allows users to better evaluate the math reasoning ability and conducts fine-grained analysis. Besides, it is a plug-and-play testing framework suitable for mathematical problems of different difficulty.
>
> We construct MathCheck-GSM for two motivations:  **(1) validate the effectiveness of MathCheck paradigm. (2) determine whether LLMs are capable of reasoning at the grade school level while most of them achieve above 85% on GSM8k before.** Through our experiments, we observe significant performance drops in many LLMs, which validates the effectiveness of our paradigm and reveals that some LLMs rely on memorization to solve GSM8k problems.
>
> Meanwhile, we also want to evaluate models on MathCheck with multimodal math problems. To address this, we developed **MathCheck-Geo comprising high-school geometry problems, on which GPT-4o achieves only 65.3% and we successfully identified weaknesses (robustness or task generalization) in some MLLMs.**
>
> **Through above datasets, the effectiveness and adaptability of MathCheck are validated.** Recently, we also recognize the community's eagerness in more challenging textual mathematical problems in MathCheck paradigm. In the future work, we will develop MathCheck with more difficult textual benchmark such as MATH. Thanks for your suggestion!
>
> > Adding Related works.
>
> Thanks for your suggestion! We indeed drew inspiration from many related works during the design of MathCheck. We have added two new sub-sections in related work(Benchmarks of Reasoning Consistency, Strategies of Improving Mathematical Reasoning) and incorporate them into new version, please check.
>
> > About the human evaluation process.
>
> Sorry for the confusion. We selected three graduate students as human annotators, all of them are not the authors. Our human evaluation principle is the generated data should maintain the correctness of mathematical logic. For example, in the “Problem Understanding”, the generated question should not alter the logical of original question, which ensures the consistency between rewritten question and answer. After that, annotators corrected each failed data instead of discarding them. This approach ensures our dataset is entirely accurate and the evaluation results are reliable. We include the detailed description of human evaluation process in the Appendix C.3, please check.
>
> Thanks for your review again! We hope our responses can address your concerns.

---

> > ### Comment · Reviewer_eZPq · 2024-11-26
> >
> > Thanks for the response and revision. I will keep my positive review.

---

> > > ### Author Response · Authors · 2024-11-26
> > > **Response to Reviewer eZPq**
> > >
> > > Dear Reriewer.
> > >
> > > Thank you for recognizing the contributions of our work. We also appreciate your insightful suggestions, which inspired us and helped improve the quality of our paper.
> > >
> > > best wishes,
> > > Authors of Paper 7051

---

### Author Response · Authors · 2024-11-21
**General response to reviewers**

Dear Reviewers,

Thanks for your review and valuable comments, which make our paper better. We have revised our paper according to your comments and uploaded it to OpenReview. In addition, for your convenience, **we have highlighted the revisions in blue in the revised paper**. Thank you again for your work and look forward to further communication with you.

Authors of Paper 7051

---

### Author Response · Authors · 2024-11-25
**General response to reviewers**

Dear Reviewers,

Thanks again for your review and we really appreciate your time and effort. We have provided detailed clarifications to your questions point by point. As the discussion period is approaching the deadline, if you have any further questions or concerns, we would be happy to discuss them with you. Thank you!


Best wishes,
Authors of Paper 7051

---

### Meta-Review · Area_Chair_PA6h · 2024-12-21

**Metareview:**

(a) Scientific Claims:
The paper introduces MathCheck, a framework for evaluating mathematical reasoning in LLMs through task generalization and reasoning robustness. Using this framework, they develop MathCheck-GSM and MathCheck-GEO for text and multimodal evaluation respectively, demonstrating that while top models maintain strong performance, many others show significant degradation compared to standard problem-solving benchmarks.

(b) Strengths:
- Novel evaluation approach beyond traditional problem-solving assessment
- Rigorous methodology with 43 models evaluated
- Careful data validation and correction process
- Strong empirical validation through correlation with private data
- Clear presentation and extensible framework

(c) Weaknesses:
- MathCheck-GSM relatively easy for frontier models
- Limited seed data types
- Initial 85% data generation pass rate
- Related work could be more comprehensive

(d) Reasons for Acceptance:
- Significant methodological contribution with comprehensive evaluation paradigm
- Strong empirical validation and careful experimental design
- Framework provides actionable insights for model improvement
- Generalizable approach applicable to other reasoning tasks
- Clear presentation and thorough documentation

**Additional Comments On Reviewer Discussion:**

The authors effectively addressed all major concerns in the rebuttal:

- Clarified their focus on comprehensive evaluation rather than just difficulty
- Provided detailed evidence for performance drop claims
- Explained data quality assurance through manual correction
- Added expanded related work sections
- Improved figures and clarified statistics
- Demonstrated specific insights about model weaknesses

All reviewers were ultimately satisfied with the responses, with reviewer n8tf upgrading their score after detailed clarifications about the framework's insights and evidence for claims. The rebuttal strengthened the paper's contributions and supported the decision to accept.

---

### Decision · Program_Chairs · 2025-01-22

Accept (Poster)